# Performance of BDS-2/3, GPS, and Galileo Time Transfer with Real-Time Single-Frequency Precise Point Positioning

**Xia Xiao [1], Fei Shen [2],\*, Xiaochun Lu [1], Pengli Shen [1] and Yulong Ge [3]**

[1] National Time Service Center, Chinese Academy of Sciences, Xi'an 710600, China; xiaoxia@ntsc.ac.cn (X.X.); luxc@ntsc.ac.cn (X.L.); shenpengli@ntsc.ac.cn (P.S.)
[2] School of Geography, Nanjing Normal University, Nanjing 210023, China
[3] School of Marine Science and Engineering, Nanjing Normal University, Nanjing 210023, China; geyulong15@mials.ucas.ac.cn
\* Correspondence: shen.f@njnu.edu.cn; Tel.: +86-0258598565

**Abstract:** Single-frequency (SF) receivers are much cheaper than that of dual-frequency (DF). Even though DF precise point positioning (PPP) is nowadays applied in the time community, the cost of equipment is one of the key considerations for time users. Furthermore, the hardware delay calibration of single-frequency devices is simpler than that of dual-frequency devices. In addition, there is no literature to study real-time SF PPP time transfer. In this contribution, the possibility of time transfer using SF PPP was studied. The Un-combined SF PPP was employed for time transfer with ionospheric-constraint using real-time precise products. In this case, 18 multi-GNSS experiment (MGEX) stations and one time lab station were used to study real-time SF PPP time transfer using GPS, Galileo and BDS-2/3 satellites with 20-day. The results suggested that real-time single-frequency PPP can meet time transfer. the standard deviation (STD) of the clock difference obtained from GPS-only, Galileo-only and BDS-2/3 single-frequency PPP are about (0.51, 0.54, 0.91) ns, respectively. The frequency stability of real-time single-frequency PPP can achieve (1E-12, 1E-13, 1E-13) level at short-term and (1E-13, 1E-13, 1E-14) level at long-term, respectively, for BDS-2/3, Galileo-only and GPS-only based.

**Keywords:** real-time; BDS-2/3; GPS; Galileo; single-frequency; time transfer; PPP





## 1. Introduction

GNSS technique is a reliable and effective tool for time transfer in the time community, due to its all-weather, high precision and other characteristics [1–5]. Even though the receiver clock offset parameter is estimated from GNSS data, GNSS has been generally regarded as one of the most accurate time transfer technologies [6,7]. Currently, GNSS time transfer techniques can be divided into two schemes as time transfer with pseudorange observations or with carrier phase observations, respectively [3,8]. In pseudorange observation-based scheme, CV, AV and TWSTFT [9] are representative time transfer technologies, with the accuracy of time transfer up to nanoseconds levels. However, the accuracy of above technologies is subject to pseudorange observations. Fortunately, in carrier phase and pseudorange observation-based scheme, PPP is one of the most accurate technologies for time transfer, which can reach sub-nanoseconds levels. Time transfer using PPP technology is not limited by the distance of the different stations and is not affected by the weather [10,11].

With the continuous upgrade of multi-GNSS, time transfer using BDS, GPS, GLONASS and Galileo PPP exhibits a hot topic, especially for BDS or Galileo PPP [5,12–15]. Nowadays, BDS-3 was successfully built since the last GEO satellite of BDS-3 was launched on 23 June 2020 [16]. 3 GEO, 24 MEO and 3 IGSO make up the BDS-3 system in current state [17,18], which enables the BDS-3 only global PNT. Furthermore, 24 Galileo satellites have provided valid navigation messages and healthy signals [19,20], which enables the Galileo-only

global PNT. GPS PPP technology was firstly applied for TAI computation in the time community since 2008 [4]. Currently, the type A uncertainty, which is the statistical uncertainty evaluate by taking into account the level of phase noise in the raw data, of PPP TAI PPP computation can reach 0.3 ns [21]. Afterword, to improve the accuracy of PPP in the time community, GLONASS, BDS, Galileo or multi-GNSS PPP time transfer was investigated by many researchers [22–26]. Ge et al. [23] presented time transfer performance with GLONASS-only PPP considering IFCBs [23]. They said that high-precision GLONASS-only PPP time transfer with estimating IFCB for each satellite performed best. Tu et al. [13] investigated the BDS-2 TF uncombined PPP model and showed that TF uncombined PPP transfer was the same as TF IF-PPP in stability and accuracy. Furthermore, PPP time transfer using Galileo observations was performed by Zhang et al. [25]. The results presented that quad-frequency PPP model can enhance the reliability and redundancy of time transfer compared to dual-frequency model. Furthermore, the performance of time transfer using multi-GNSS was indicated by Zhang et al. [27] and Ge et al. [5]. In addition, frequency transfer with integer ambiguity PPP techniques was further realized [28,29]. IPPP can realize frequency transfer with the level of sub -10-16/T in the long-term stability, where T is the duration in days of continuous phase measurements.

The previous studies mainly study time transfer in post-processing PPP model, which isn't employed for real-time time transfer. For the above background, researchers have further studied real-time time transfer with PPP technology. Monitoring of UTC(k)'s was investigated by Defraigne et al. [30] using PPP technology with IGS ultra-rapid products. Li et al. [31] realized time transfer with sub-nanosecond levels using real-time PPP technology. In addition, time transfer in post processing PPP has a daily boundary discontinuous. Luckily, real-time time transfer can solve the daily boundary discontinuous problem perfectly. More interestingly, Ge et al. [32] studied the real-time time transfer considering the receiver clock offset model in PPP. The conclusions presented that the performance of time transfer with PPP technology was enhanced obviously by the clock model. Then, the clock model was applied to time transfer during date discontinuity by Qin et al. [33]. The results showed that the accuracy of time transfer could be improved obviously by using the clock model. In addition, based on real-time PPP technology, the reference time of precise clock products was set to UTC(k) [34,35] or GNSS time in broadcast ephemeris [12], precise timing was then implemented. Here, k refers to time lab. Dual- or multi-frequency observations was used by the above article of PPP in the time community. Currently, equipment cost is one of the most important factors for current GNSS users [36]. Actually, single-frequency PPP in precise positioning has been studied by many researchers [37,38]. Single-frequency time transfer in the post-processing PPP was exhibited by Ge et al. [39]. Their results indicated that single-frequency PPP time transfer with ionospheric-constraint performs better than other two single-frequency PPP model (ionospheric-free and ionospheric-corrected single-frequency PPP model). However, the research on real-time time transfer is still limited with single-frequency PPP. Hence, our work is devoted to the study of real-time time transfer using single-frequency PPP with multi-GNSS observations. According to the conclusions obtained from Ge et al. [39], the single-frequency PPP technology with ionospheric- constraint was employed to time transfer.

We first presented the basic technology of single-frequency PPP and described the flow of real-time time transfer with single-frequency PPP. The accuracy of products from CNES were assessed. Following that the datasets and processing strategies were given in detail. Afterwards, real-time time transfer with single-frequency PPP with BDS-2/3, GPS, and Galileo was presented and analyzed. Finally, the conclusions were presented in our work.

## 2. Methods

The function and stochastic model of single-frequency PPP time transfer was outlined firstly. Then, an introduction of real-time time transfer method using single-frequency PPP was described in detail.

### 2.1. Observations

The single-frequency uncombined observations can be described as [40,41]:

$$\begin{cases} p_{r,i}^s = e_r^s \cdot \Delta x + cdt_{r,i} + M_r^s \cdot Z_w + I_{r,i}^s + \beta_{ij} \cdot DCB_{ij}^s + cd_{r,i} + \varsigma_{r,i}^s \\ l_{r,i}^s = e_r^s \cdot \Delta x + cdt_{r,i} + M_r^s \cdot Z_w - I_{r,i}^s - cd_{IF_{ij}}^s + \lambda_1(N_{r,i}^s + b_{r,i} + b_i^s) + \varsigma_{r,1}^s \end{cases} \quad (1)$$

$$\begin{cases} \alpha_{ij} = \frac{(f_i^s)^2}{(f_i^s)^2 - (f_j^s)^2} \\ \beta_{ij} = -\frac{(f_j^s)^2}{(f_i^s)^2 - (f_j^s)^2} \\ DCB_{ij}^s = d_i^s - d_j^s \\ d_{IF_{ij}}^s = \alpha_{ij} \cdot d_i^s + \beta_{ij} \cdot d_j^s \end{cases} \quad (2)$$

where $l$ and $p$ indicate the OMC values of carrier phase and pseudorange observables; $e_r^s$ is the unit vector of the component from the receiver to the satellites; $c$ is the speed of light; $r$ and $s$ refer to the used receiver and satellite; The precise satellite clock correction is the sum of satellite clock offset and a specific linear function of the satellite UCDs. When the precise satellite clock was applied for single-frequency PPP, the DCB at satellite end ($DCB_{ij}^s$) should be corrected; Here, $I$ and $j$ denote the frequency; $\Delta x$ is the coordinate increments in 3D components in meters; $dt_{r,i}$ represents the receiver clock; $M_r^s$ present the mapping function; $Z_w$ is the ZWD; $I_{r,i}^s$ illustrates the slant ionospheric delay; $\beta_{ij}$ are the frequency factors ($i \neq j$). $d_{r,i}$ refer to the UCD at receiver end, respectively; $d_{IF_{ij}}^s$ is the satellite UCDs; $d_i^s$ and $d_j^s$ are the satellite UCD at frequency $i$ and $j$; $\lambda_i$ indicates the wavelength; $N_{r,i}^s$ represents the integer ambiguity; $b_{r,j}$ and $b_j^s$ are phase delay. $\xi_{r,j}^s$ and $\varsigma_{r,j}^s$ are the noise.

Obviously, Equations (1) and (2) present a rank-deficient. To solve the ionospheric delay parameterization in single-frequency PPP, the DESIGN model was adopted in our work [42], as:

$$I(z)_r^s = A_0 + A_1 dL + A_2 dL^2 + A_3 dB + A_4 dB^2 + R_r^s + \zeta_r^s \quad (3)$$

where $dB$ and $dL$ are the latitude and longitude difference between the coordinate of station and IPP. $A_0$ is the mean value of ionospheric delay; $A_1$, $A_2$, $A_3$ and $A_4$ refer to the corresponding coefficients, respectively; $dL$ and $dB$ are the longitude and latitude difference between the ionospheric IPP and the approximate location of station, respectively; $R_r^s$ indicates a random part; $I(z)_r^s$ illustrates the vertical ionospheric delay correction interpolated from GIM or an available regional ionosphere model with corresponding noise $\zeta_r^s$.

Considering the accuracy of GIM products, the virtual ionospheric observations were given more weight at the beginning of PPP to achieve rapid convergence, but their weight is gradually reduced after convergence. Hence, the progressive relaxation constraint was employed and can be written as [43]:

$$\sigma_{\varepsilon_{r,ion}^s}^2(i) = \sigma_{\varepsilon_{ion},0}^2 + \alpha(i-1)\Delta t \quad (4)$$

where $\alpha$ is the rate of change of variance (m$^2$/min); $\Delta t$ illustrates the sample interval of observations; $\sigma_{\varepsilon_{ion},0}^2$ and $\alpha$ can be set as 0.09 m$^2$ and 0.04 m$^2$/min [43].

### 2.2. Time Transfer with Real-Time Single-Frequency PPP

A flowchart of real-time single-frequency PPP time transfer processing procedure is presented in Figure 1. One station, namely $A$, located in the timing lab, was connected to their frequency generator. We can obtain the observations from the station and broadcast ephemeris, SSR corrections and ionospheric DCB products from IGS via internet, firstly. Then, the receiver clock offset can be estimated by single-frequency PPP model. For the

time-links (including station $A$ and $B$ located in different place), we can obtain the receiver clock offset as:

$$T_A = t_A - t_{ref} \qquad (5)$$

$$T_B = t_B - t_{ref} \qquad (6)$$

where $T_A$ and $T_B$ are the receiver clock offset for station $A$ and $B$; $t_A$ and $t_B$ denote the local time, respectively. $t_{ref}$ is the reference of the real-time products.

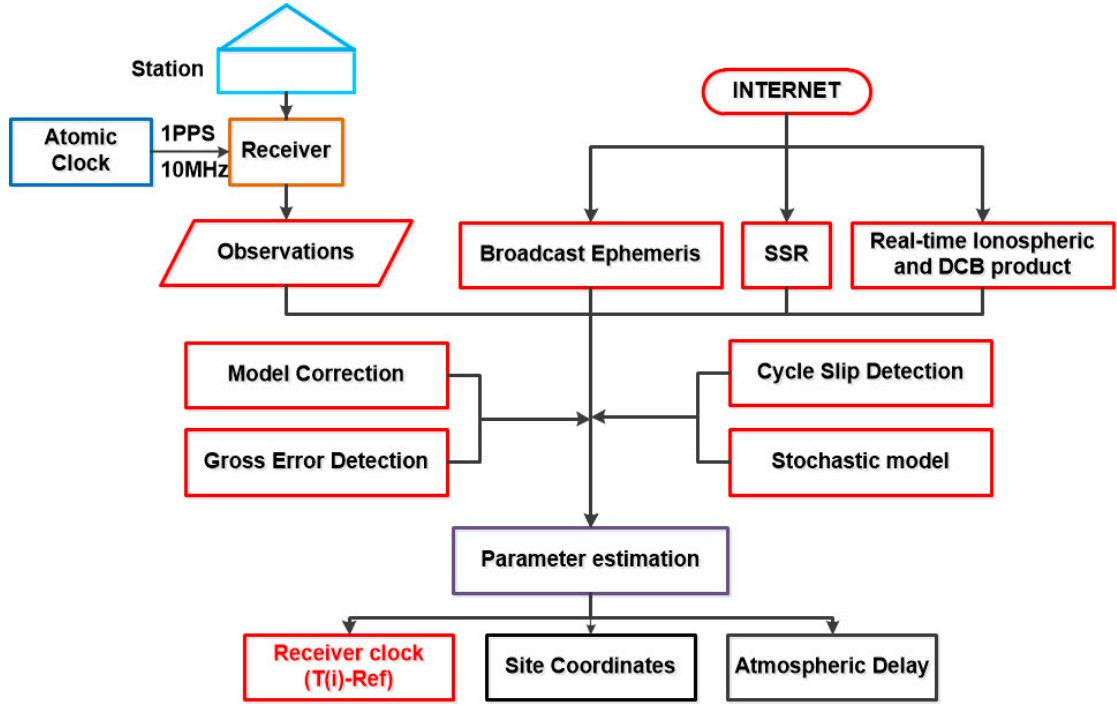

**Figure 1.** The flowchart of real-time time transfer with single-frequency PPP. Note that the site coordinates can be fixed. If the site coordinates are estimated as parameter in static model, the receiver clock should be used after the convergence.

Then, the time difference, namely $\Delta T$, can be calculated by the Equation (7) via internet as:

$$\Delta T = T_B - T_A \qquad (7)$$

## 3. Accuracy of Real-Time Precise Product

The accuracy of precise products directly determines the performance of real-time PPP time transfer. To present the performance of real-time time transfer with single-frequency PPP using BDS-2/3, GPS, and Galileo, the assessment of real-time precise products is firstly introduced. Here, the final products released from GFZ from DOY 30 to 50, 2021 were employed to evaluate the accuracy of real-time precise products released by CNES. The average RMSs of the orbit errors for GPS, Galileo and BDS-2/3 are displayed in Figures 2 and 3 at RAC, respectively. It is Noted that not all BDS-3 satellites are contained in CNES's products at present, only the satellites presented in the Figure 3. From the figures, we can find that the mean RMSs of orbits errors of GPS and Galileo are (0.026, 0.038, 0.032) m and (0.038, 0.080, 0.051) m at RAC direction. In addition, the mean RMSs of orbits for BDS-2 GEO, IGSO/MEO and BDS-3 are (0.35, 3.73, 1.84) m, (0.16, 0.32, 0.19) m and (0.11, 0.28, 0.16) m at RAC direction. In addition, the accuracy of BDS-3 outperforms that of BDS-2 IGSO/MEO. The accuracy of BDS orbit in real-time precise products still has a large room for improvement.

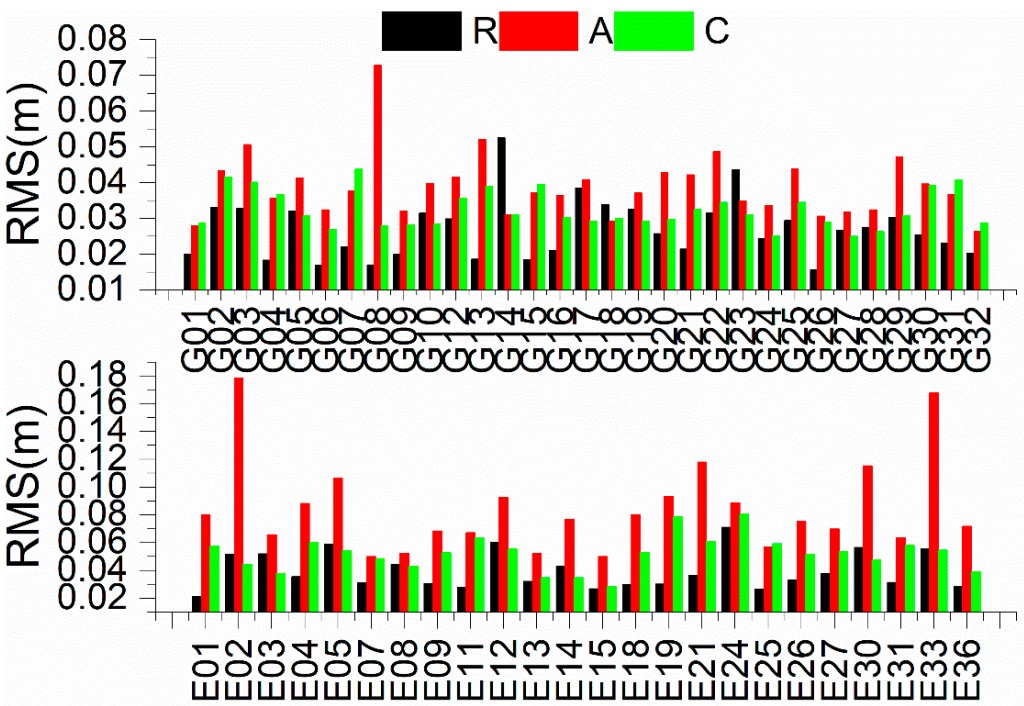

**Figure 2.** The RMS of orbits error for GPS and Galileo with compared to GFZ final products at RAC direction.

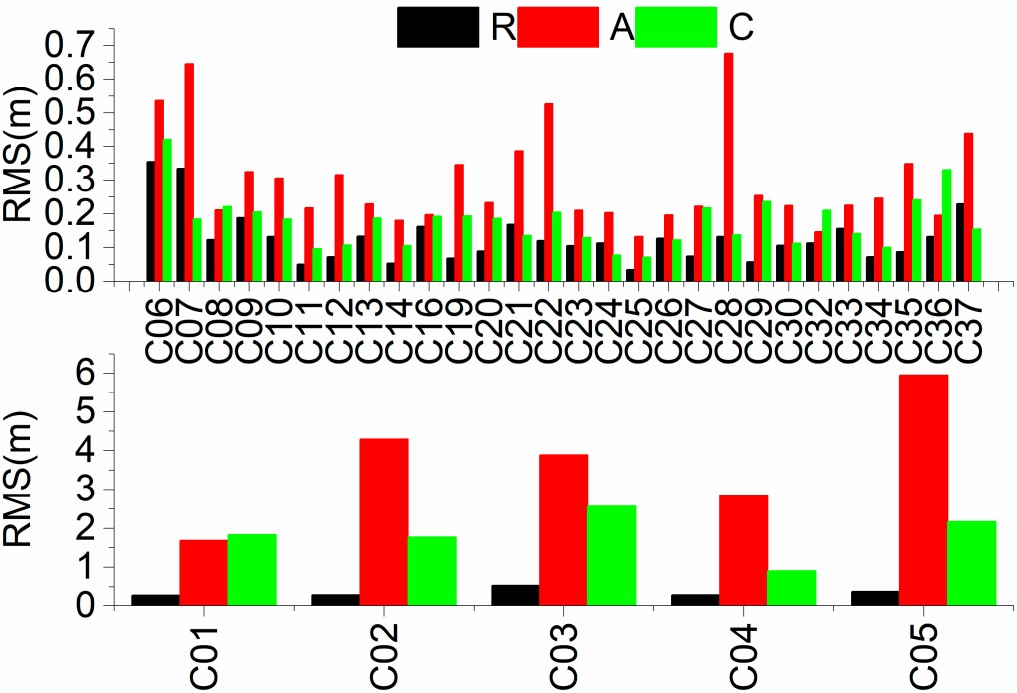

**Figure 3.** The RMS of BDS-2/3 satellite orbits errors with compared to GFZ final products at RAC direction.

Additionally, the STDs of satellites clock offset difference for GPS, Galileo and BDS-2/3 are presented in Figure 4. We see that STDs of the difference for GPS and Galileo are mainly less than 0.2 ns. The STDs of the difference are better than (1.1, 0.6, 0.6) ns for BDS-2 GEO, IGSO/MEO and BDS-3, respectively. The mean STD values are (0.18, 0.16) ns for GPS/Galileo, respectively, which is similar to the results as the paper [44]. Furthermore,

the mean STD values for BDS-2 GEO, IGSO/MEO and BDS-3 satellites are (0.9, 0.58, 0.51) ns, respectively. The findings agree with the above conclusions.

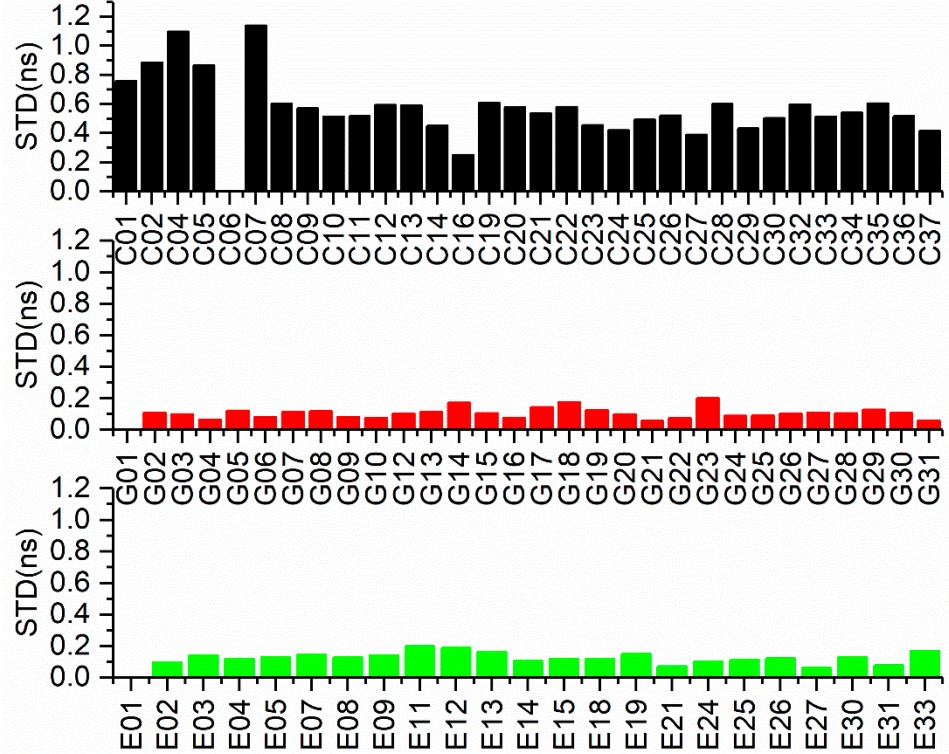

**Figure 4.** The STD of the clock difference for GPS, Galileo and BDS-2/3 satellites with compared to GFZ final products.

## 4. Test Data and Processing Strategies

Stations selected in our work are introduced firstly. Then, we introduce the processing strategies for real-time time transfer with single-frequency PPP in this contribution.

### 4.1. Dataset

To investigate BDS-2/3-, GPS- and Galileo-only time transfer with single-frequency PPP, 18 stations from MGEX or one station from timing lab were selected and presented in Figure 5. which covered 20 days from DOY 30 to 50, 2021. PTBB was set as center node. The detailed information of the selected stations is listed in Table 1 (such as receiver, antenna and atomic clock). The selected stations are all connected to the high-precise atomic clock. Then, 18 time-links were designed to study real-time time transfer with single-frequency PPP. BNC software was applied for receiving precise products released by CNES and the observations. Then, real-time time transfer performance using single-frequency PPP was given using a secondary development of GAMP software [45]. Additionally, IGS final clock products were set as the true values to assess real-time time transfer. Time transfer solutions is to obtain the time difference between different stations.

**Table 1.** Details in formation of the selected IGS stations.

| Station | Receiver | Antenna | Clock |
| --- | --- | --- | --- |
| PTBB | SEPT POLARX4TR | LEIAR25.R4 LEIT | UTC(PTB) |
| BRUX | SEPT POLARX4TR | JAVRINGANT_DM | UTC(ROB) |
| PTBB | SEPT POLARX4TR | - | UTC(PTB) |
| AREG | TRIMBLE NETR9 | TRM59800.00 | RUBIDIUM |
| CEBR | SEPT POLARX4 | SEPCHOKE_MC | H-MASER |

**Table 1.** *Cont.*

| Station | Receiver | Antenna | Clock |
|---------|----------|---------|-------|
| KIRU | SEPT POLARX4 | SEPCHOKE_MC | CESIUM |
| KOUR | SEPT POLARX4 | SEPCHOKE_MC | H-MASER |
| MAS1 | SEPT POLARX4 | LEIAR25.R4 | CESIUM |
| ONSA | JAVAD TRE_G3TH DELTA | AOAD/M_B | H-MASER |
| PIE1 | JAVAD TRE_G3TH DELTA | ASH701945E_M | H-MASER |
| REDU | SEPT POLARX4 | SEPCHOKE_MC | CESIUM |
| SCOR | JAVAD TRE_G3TH SIGMA | ASH701941.B | RUBIDIUM |
| SPT0 | JAVAD TRE_G3TH DELTA | JNSCR_C146-22-1 | H-MASER |
| VILL | SEPT POLARX4 | SEPCHOKE_MC | CESIUM |
| YEL2 | SEPT POLARX4TR | LEIAR25.R4 | H-MASER |
| USN7 | ASHTECH Z-XII3T | TPSCR.G5 | H-MASER |
| ROAG | SEPT POLARX4 | LEIAR25.R4 NONE | H-MASER |
| HOB2 | SEPT POLARX5 | AOAD/M_T | H-MASER |
| HARB | TRIMBLE NETR9 | TRM59800.00 | CESIUM |

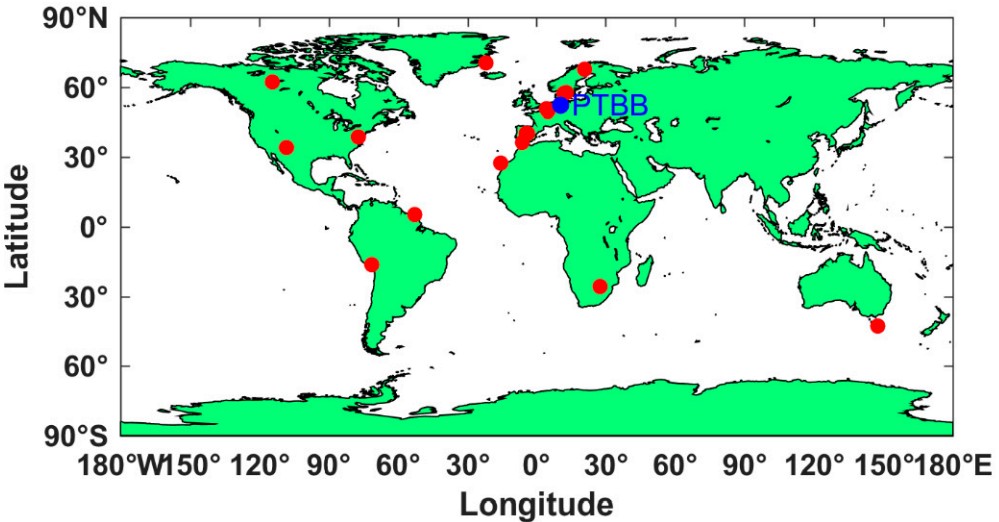

**Figure 5.** The location of 18 MGEX stations and one station located in PTB (namely PT11). All the stations are equipped with high-precise atomic clock.

### 4.2. Processing Strategies

For real-time single-frequency PPP, the phase windup, relativistic effects and tidal loadings et al. were corrected by the models [46]. The PCOs and PCVs were corrected by the atx file. Table 2 gives the processing strategies in detail.

**Table 2.** Information of processing strategies.

| Items | Models |
|-------|--------|
| Estimate | Kalman filter |
| Cutoff angle | 10° |
| Signal selection | BDS-2/BDS-3: B1I; Galileo: E1; GPS: L1 |
| Sampling rate | 30 s |
| Phase wind-up | Corrected [47] |
| Tropospheric delay | ZHD: corrected by models [48] |
| | ZWD: estimated using GMF [49] mapping function |
| Tidal displacement | Corrected [46] |

**Table 2.** *Cont.*

| Items | Models |
|---|---|
| Sagnac effect | Corrected [46] |
| Relativistic effect | Corrected [46] |
| PCO and PCV | Corrected |
| Phase ambiguities | Estimate as constant |
| Receiver clock offset | White noise |
| Station coordinates | Estimated |
| ISB between BDS-2 and BDS-3 | Estimated as white noise |

## 5. Results

In this subsection, the real-time time transfer using GPS, Galileo and BDS-2/3 single-frequency PPP was tested and investigated. The characteristic of real-time time transfer with single-frequency PPP is presented. Note that the time transfer results from IGS final clock products were regarded as the reference.

### 5.1. GPS

The receiver clock offsets of PTBB, BRUX, PT11, ROAG, REDU and VILL from GPS real-time single-frequency PPP are displayed in Figures 6–8. Here, we randomly select the stations, other stations show similar performance, hence, we will not show them in detail. Note that, as we mentioned previous, the receiver clock offset indicates the difference between the clock reference difference of the real-time precise products and the local time. From the figures, three findings are presented. First, obviously, the clock offset is relatively stable for all stations before DOY 43, 2021, while there is a significantly larger fluctuation after DOY 43, 2021. That fact may be presented that the clock reference difference of real-time products released by CNES is always changing, and the varies values is very large [34]. The epoch-wise average over satellites (clock reference difference), which should vary very much from epoch to epoch in the 2nd half time. In generally, the clock in the timing lab is very stable for short periods of time. Furthermore, the above fact also leads to the reason that it cannot implement precise timing [34,50]. We can prove this fact from the figure at right side in Figure 6, exhibiting the receiver offset obtained from single-frequency PPP with IGS final products. The time series from IGS final products presents a stable characteristic. Second, there was some missing data that caused the receiver clock offset of PT11 displayed in Figure 7 to interrupt. Note that PT11 was located at PTB and connect with high-precise clock. Third, compare Figures 6–8, we can see that the clock offset of PTBB, BRUX, PT11 and ROAG are more stable than that of REDU and VILL. This reason is that the performance of atomic clock connects to PTBB, BRUX, PT11 and ROAG is better than that of REDU and VILL.

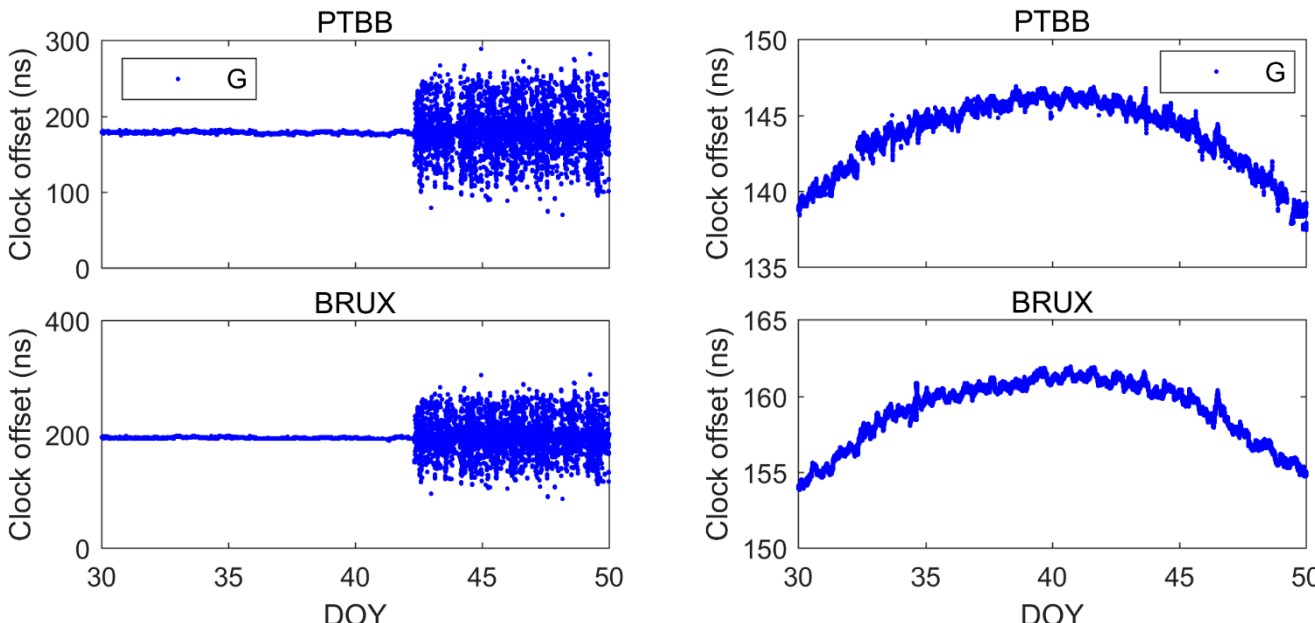

**Figure 6.** Clock offsets of PTBB and BRUX from GPS-only single-frequency PPP. Here, G refer to GPS satellites. The figure in the left and in the right side were obtained by using the real-time products and IGS final products, respectively.

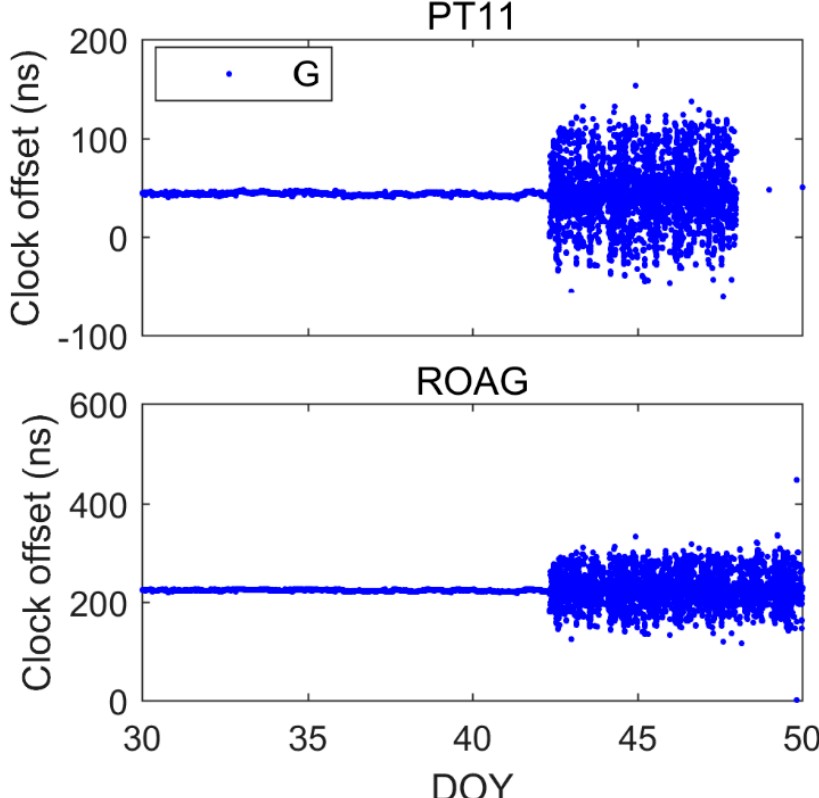

**Figure 7.** Clock offsets of PT11 and ROAG from GPS-only single-frequency PPP. Here, G refer to GPS satellites.

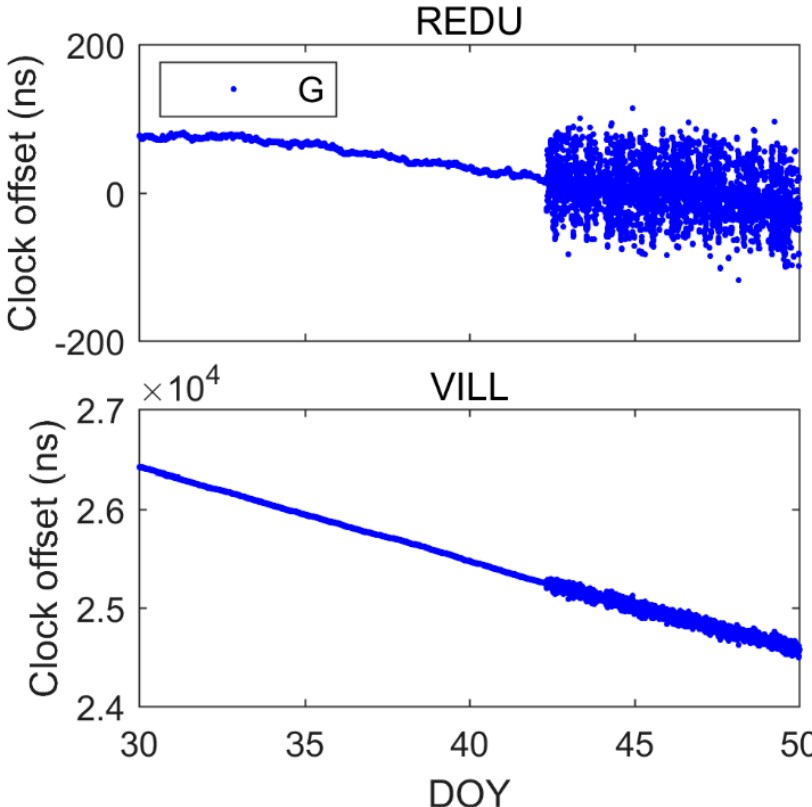

**Figure 8.** Clock offsets of REDU and VILL from GPS-only single-frequency PPP. Here, G refer to GPS satellites.

Figure 9 displays the time transfer solutions of PT11-, BRUX-, ROAG- and REDU-PTBB time-links from GPS-only single-frequency PPP with real-time precise products. It is of interest that PTBB and PT11 are equipped with one atomic clock. The clock difference of PT11-PTBB time-link only includes the noise and the hardware delay. The values of hardware delay can be considered as a constant in the short term. Hence, the level of noise for PT11-PTBB can reflect the accuracy of single-frequency PPP time transfer with real-time products. From Figure 9, interestingly, the time series of PT11-PTBB exhibit noise characteristics and there is no obvious linear trend. Hence, we can conclude that real-time single-frequency PPP is suitable for application to time transfer. Furthermore, the series of PT11-, BRUX-, ROAG- and REDU-PTBB present an obvious system difference. That's because we didn't calibrate the hardware delay. Since the hardware delay calibration is rather complicated, this article does not focus on the hardware delay calibration. Moreover, the tendency of PT11-, BRUX- and ROAG-PTBB is stable, while that of REDU-PTBB presents a linear trend. This trend is determined by the characteristics of the atomic clock. To further investigate real-time time transfer with GPS-only single-frequency PPP, the mean and STD values of the clock difference are exhibited in Figure 10. Two findings are presented. First, the mean values of the difference are not equal to zeros, and the mean values of the difference for different time-links are not equal. The system difference is the difference between time transfer solutions from single-frequency PPP and IGS final clock product. These problems can be explained from two aspects. One is that the time transfer in IGS final products is based on DF ionospheric-free. The DF ionospheric-free hardware delay is absorbed by the receiver clock offset. However, only hardware delay on first frequency is absorbed by the receiver clock offset from single-frequency PPP. Another is that the hardware delay is not the same at different station. Second, the STD values of the difference range from 0.18 to 0.94 ns for all time-links and the average STD value is 0.51 ns. It can further prove our previous conclusion that GPS-only single-frequency PPP with real-time products can be employed for time transfer with Type A uncertainty of about 0.51 ns.

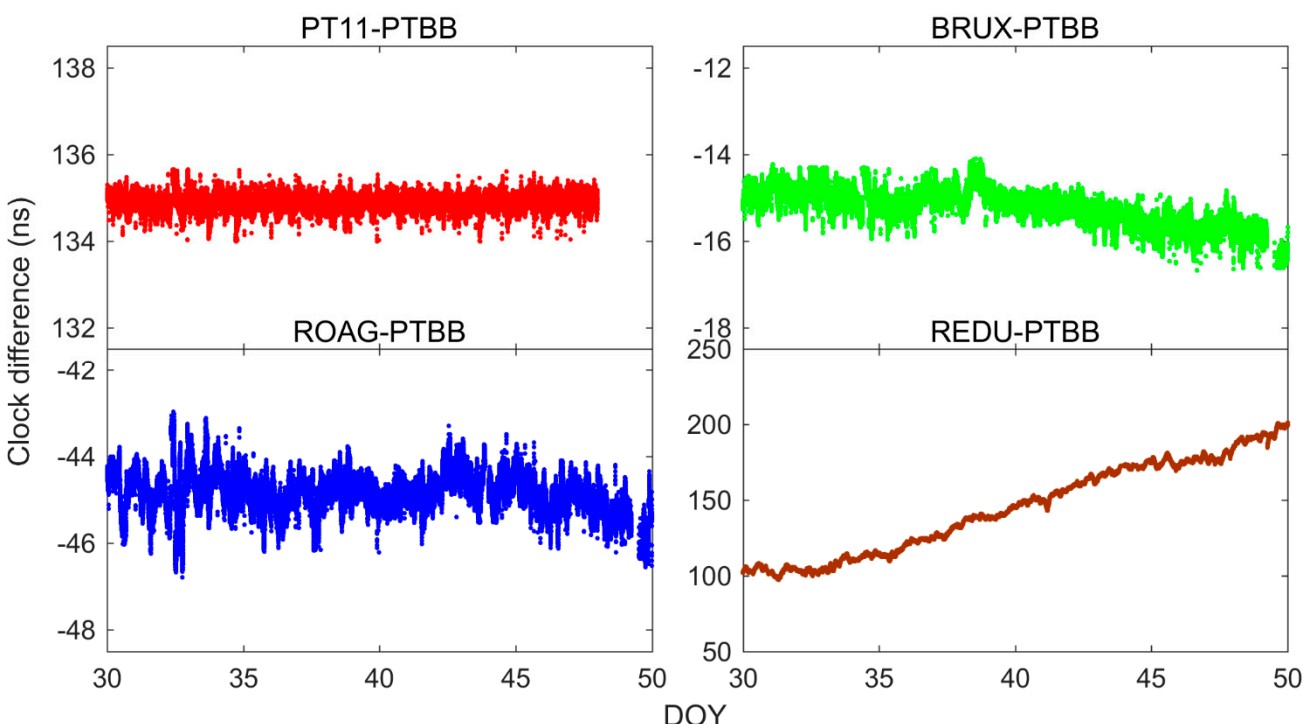

**Figure 9.** Clock difference of PT11-, ROAG-, BRUX- and REDU-PTBB from GPS-only single-frequency PPP.

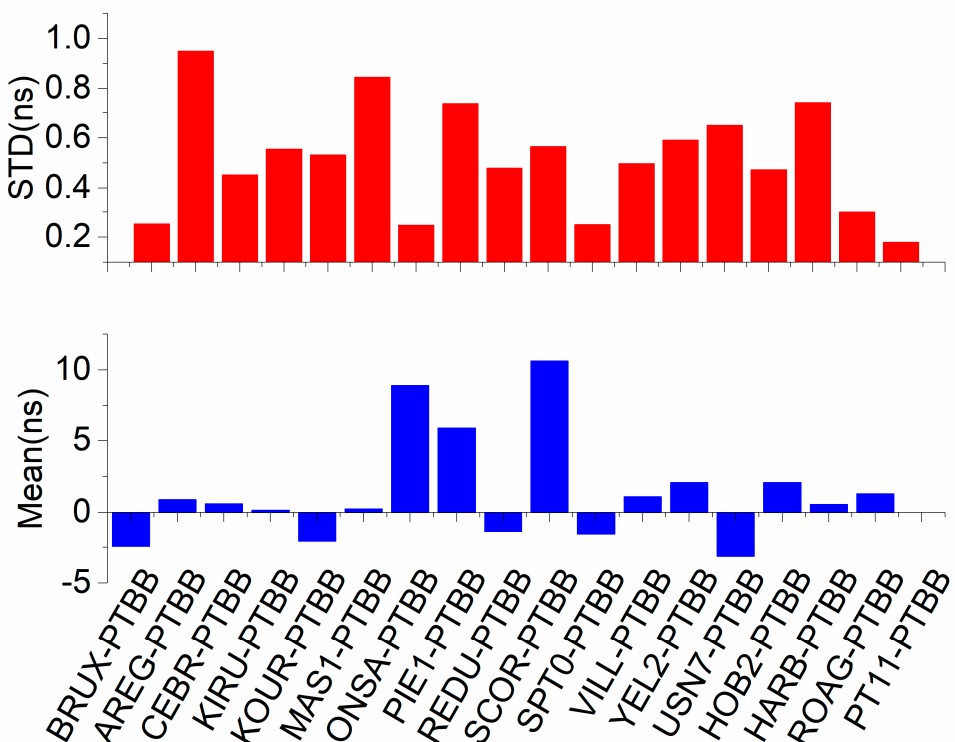

**Figure 10.** The mean and STD values of the clock difference of 18 time-links calculated from GPS-only single-frequency PPP. Note that the clock difference refers to the difference between time transfer solutions from single-frequency PPP and time transfer solutions from IGS final clock products. The.

MDEVs of 11 time-links equipped with H-master atomic and 7 time-links equipped with Rubidium/Cesium atomic clock are presented in Figures 11 and 12, respectively. From two figures, the frequency stability in Figure 11 outperform that of time-links in



Figure 12. Note that Tau is the sampling period, the same below. The frequency stability also is determined by the characteristic of atomic clock, which is the same conclusions as we previous pointed out. The stability of 11 time-links in Figure 11 is (1.4273E-13, 2.5183E-13, 3.3167E-13, 1.5535E-13, 2.2665E-13, 1.9345E-13, 2.4737E-13, 2.05917E-13, 3.4055E-13, 2.0386E-13, 1.5116E-13) at 960 s, respectively, and is (4.1521E-14, 1.0019E-13, 9.3432E-14, 5.3851E-14, 6.134E-14, 5.8668E-14, 9.5624E-14, 6.8272E-14, 1.9096E-13, 6.3041E-14, 4.0842E-14) at 15,360 s, respectively. In addition, the frequency stability of 7 time-links in Figure 12 is (1.8614E-12, 9.9491E-13, 1.9875E-12, 3.616E-13, 1.8298E-12, 3.1481E-13, 6.0605E-13) at 960 s, respectively, and is (3.8233E-13, 4.5373E-13, 5.332E-13, 1.8825E-13, 3.9287E-13, 1.4482E-13, 1.5671E-13) at 15,360 s, respectively. Hence, one conclusion is presented that the frequency stability of GPS-only single-frequency PPP can reach the level of 1E-13 and 1E-14 at short- and long- term stability.

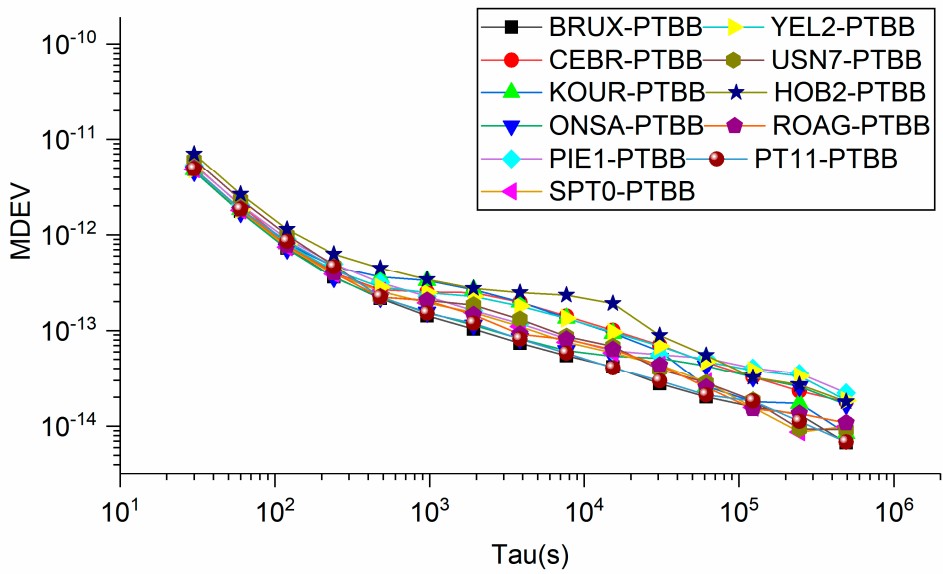

**Figure 11.** MDEV of the time-links from GPS-only single-frequency PPP For the 11 stations are equipped with H-master clock. Tau is the sampling period.

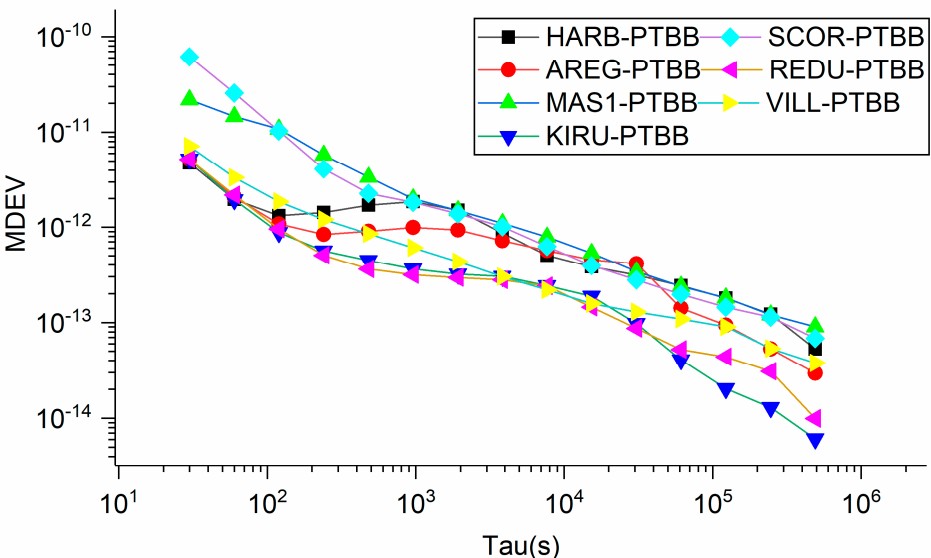

**Figure 12.** MDEV of the time-links obtained from GPS-only single-frequency PPP For the five stations equipped with Rubidium and Cesium atomic clock.

*5.2. Galileo*

Figures 13–15 indicate the receiver clock offset of PTBB, BRUX, PT11, ROAG, REDU and VILL obtained from Galileo-only single-frequency PPP with real-time product. In addition, clock differences of PT11-, BRUX-, ROAG- and REDU-PTBB time-links calculated from Galileo-only single-frequency PPP are display in Figure 16. Combined Figures 13–16, two findings are of interest suggested. First, similar to the result of GPS-only, the time series present an obvious fluctuation after DOY 43, 2021. The reason has been described in the previous, so we don't introduce it in detail here. Second, the tendency of clock difference in Figure 16 is consistent with that of GPS-only, especially for the time series of PT11-PTBB. It further proves the reliability and feasibility of real-time time transfer with Galileo-only single-frequency PPP. To further quantify our conclusions, the STD and mean values of clock difference for 18 time-links are presented in Figure 17. Similar to GPS, the solutions of real-time time transfer using Galileo-only single-frequency PPP present a system bias with IGS final products. The STD of clock difference are 0.2–0.89 ns and the average STD values reach 0.54 ns. Here, we obtained that the accuracy of real-time time transfer using Galileo-only single-frequency PPP can achieve sub-nanosecond level.

Figures 18 and 19 present the MDEV of 11 time-links equipped H-master atomic clock and 7 time-links equipped with Rubidium/Cesium atomic clock. Combined Figures 11, 12, 18 and 19, it can be noted that the frequency stability of Galileo-only is comparable to that of GPS in the short term, while the frequency stability of GPS-only presents better performance than that of Galileo-only in the long-term. That's because Galileo has relatively fewer satellites than GPS and the accuracy of real-time precise products is worse than that of GPS (see Figures 2 and 4). The frequency stability in Figure 18 is (1.3758E-13, 1.814E-13, 2.5505E-13, 1.5754E-13, 2.1162E-13, 1.5424E-13, 1.8492E-13, 1.6134E-13, 3.2023E-13, 2.3958E-13, 2.0758E-13) at 960 s, respectively, and is (1.191E-13, 4.5314E-13, 3.8586E-13, 3.7122E-13, 5.9243E-13, 4.5252E-13, 1.7635E-13, 2.0434E-13, 2.8784E-13, 1.291E-13, 1.191E-13) at 15,360 s, respectively. Hence, we suggest that the performance of real-time time transfer with Galileo-only single-frequency PPP is slightly poor that that of GPS. The frequency stability of real-time transfer with Galileo-only single-frequency PPP achieves 1E-13 level at short- and long-term.

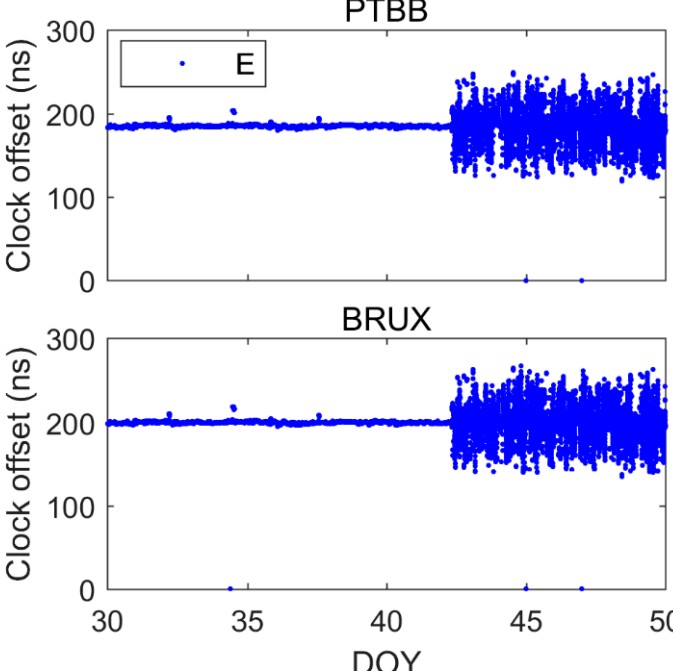

**Figure 13.** Clock offsets of BRUX and PTBB from Galileo-only single-frequency PPP. Here, E is Galileo satellites.

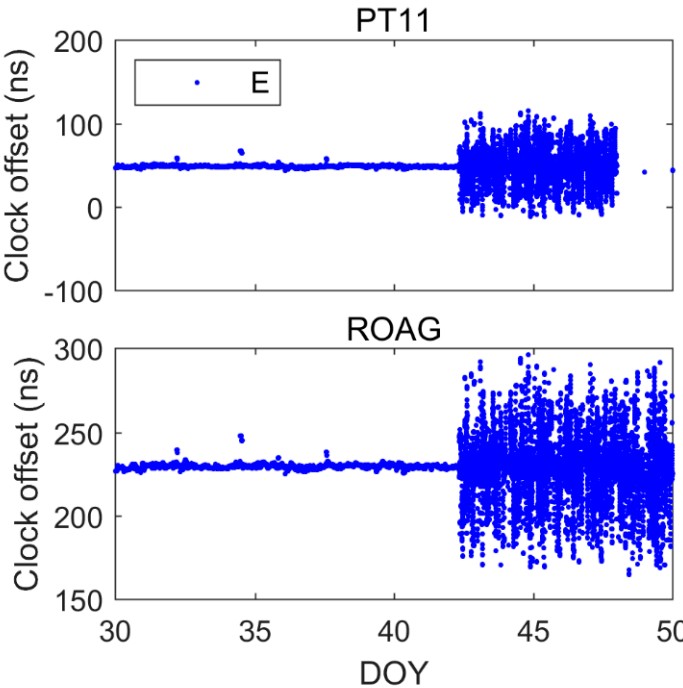

**Figure 14.** Clock offsets of ROAG and PT11 from Galileo-only single-frequency PPP. E indicates Galileo satellites.

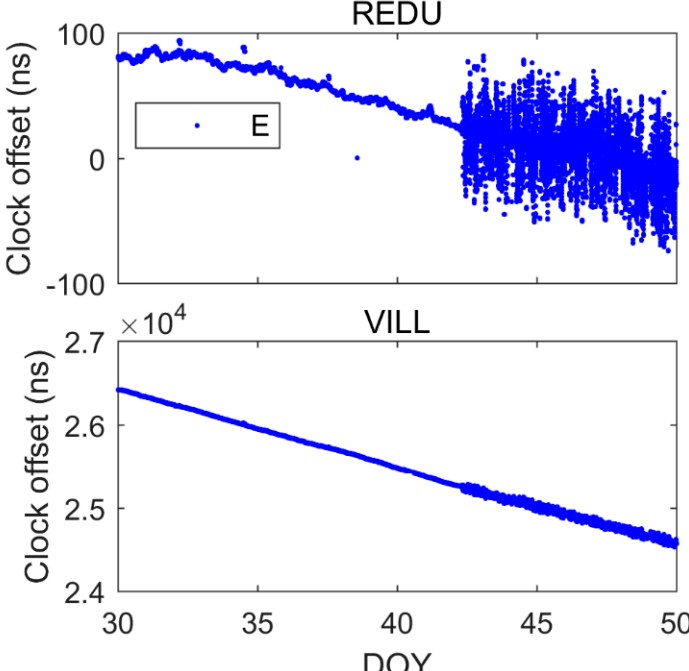

**Figure 15.** Clock offsets of REDU and VILL stations obtained from Galileo-only single-frequency PPP.

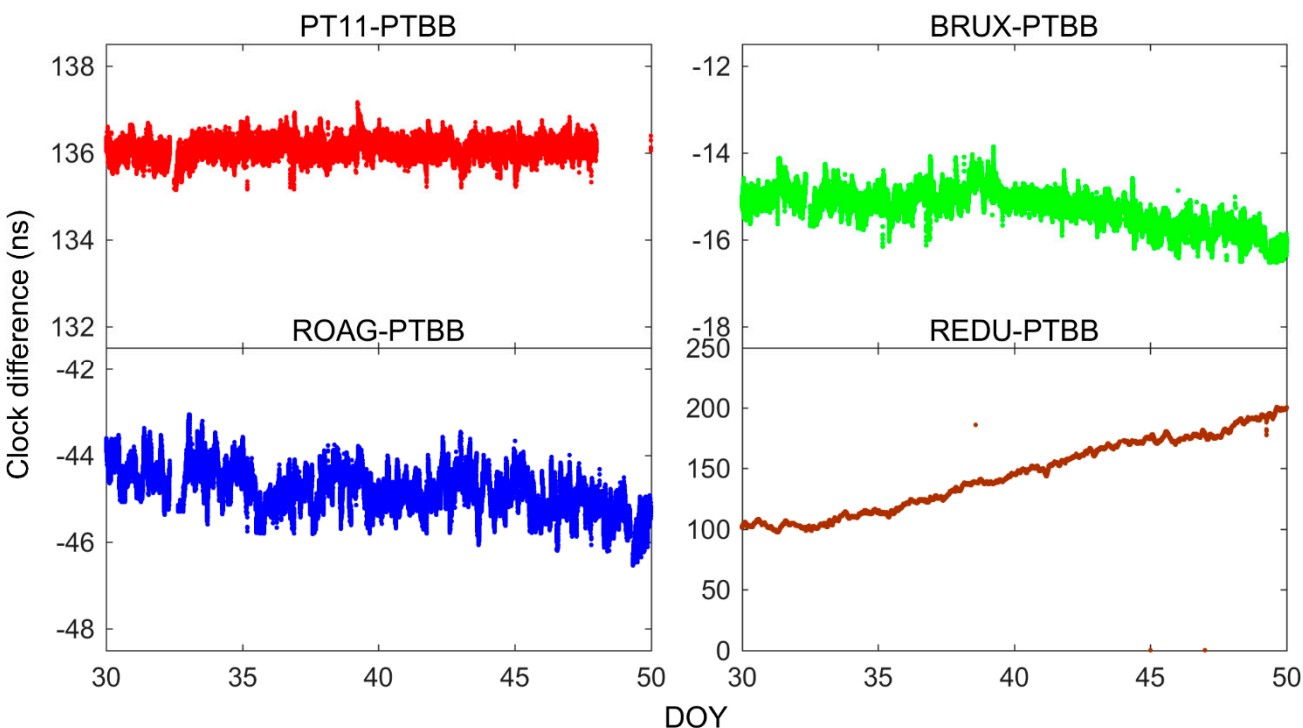

**Figure 16.** Clock difference of PT11-, ROAG-, BRUX- and REDU-PTBB time-links from Galileo-only single-frequency PPP.

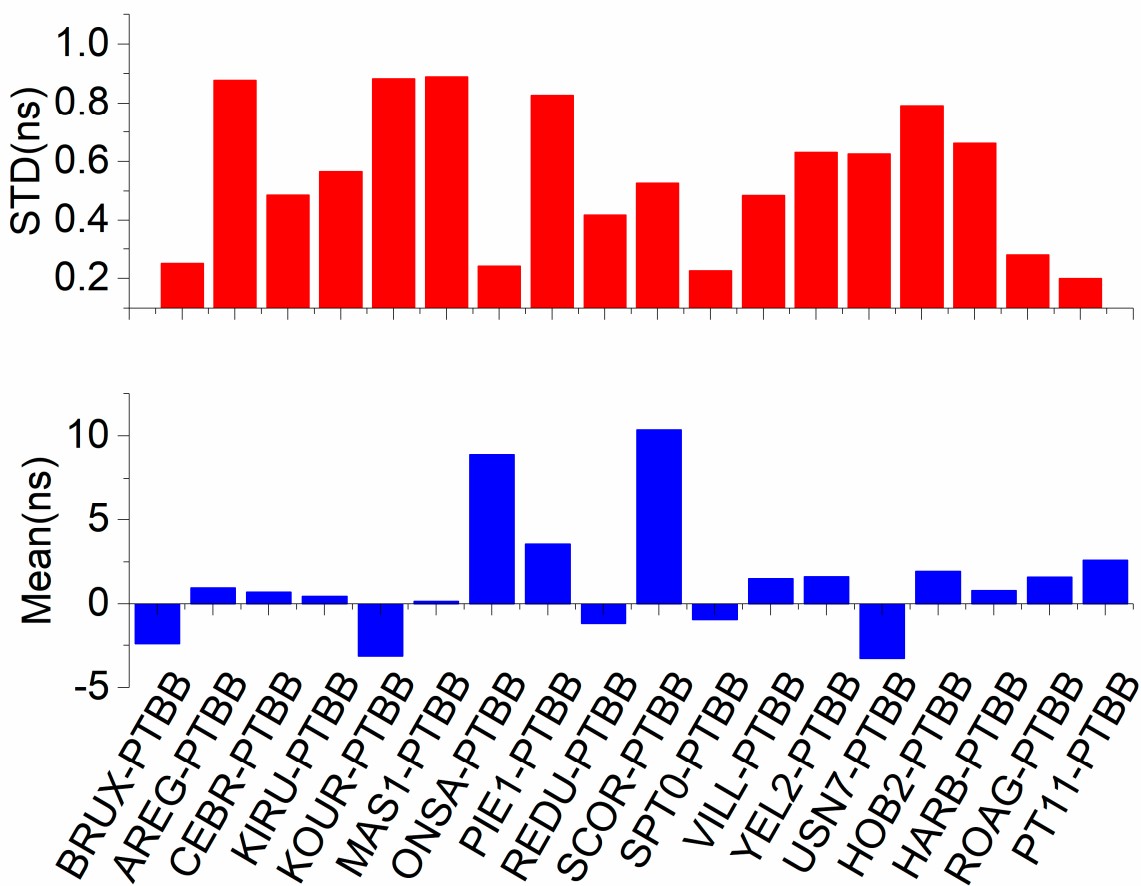

**Figure 17.** The mean and STD values of the clock difference of 18 time-links calculated from Galileo-only single-frequency PPP. Note that the clock difference refers to the difference between time transfer solutions from single-frequency PPP and time transfer solutions from IGS final clock products.

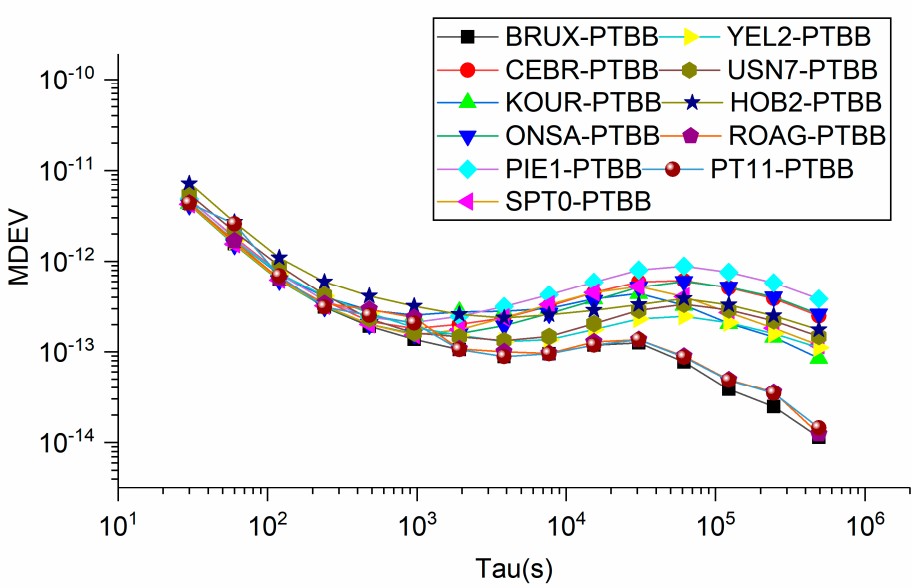

**Figure 18.** MDEV of the time links obtained from Galileo-only single-frequency PPP for the 11 stations equipped with H-master clock.

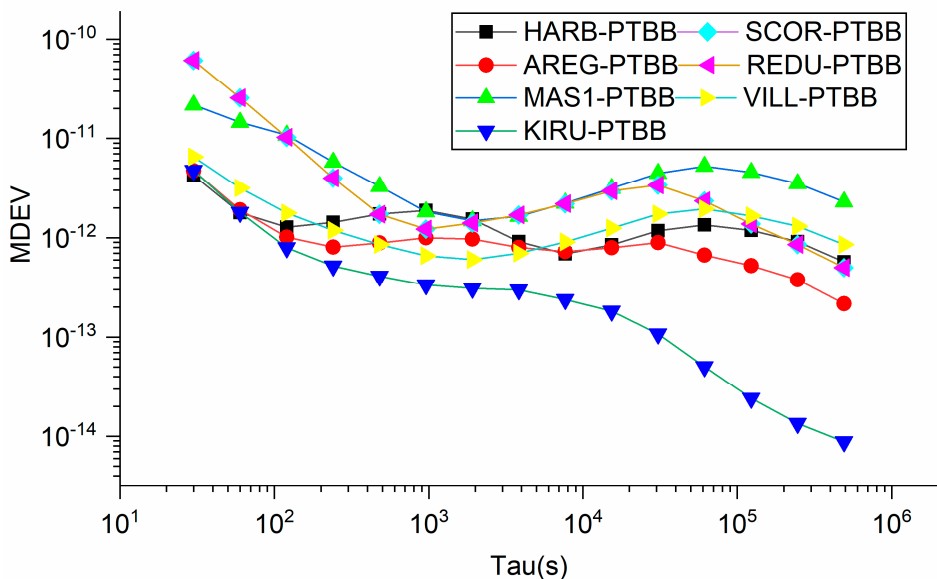

**Figure 19.** MDEV of the time links obtained from Galileo-only single-frequency PPP for the 7 stations equipped with Rubidium and Cesium atomic clock.

*5.3. BDS-2/3*

The time series of receiver clock offset for PTBB, BRUX, REDU and VILL stations are presented in Figures 20 and 21 obtained from BDS-2/3 single-frequency PPP with real-time precise products. In addition, the clock differences for VILL-, BRUX-, SPT0- and REDU-PTBB obtained from BDS-2/3 single-frequency PPP with real-time precise products are introduced in Figure 22. In the figures, part of the time series for PTBB was interrupted due to missing data from BDS-3 during that time, but that does not affect our conclusions. From three figures, three findings are presented. First, the time series obtained from BDS-2/3 single-frequency PPP show a slight parabolic behavior. That may be affected by the reference of BDS-2/3 satellite clock (see Figures 20 and 21). Second, the time series of the solutions from BDS-2/3 exhibit a similar tendency about that of GPS- and Galileo-only, which confirm the feasibility of real-time time transfer using BDS-2/3 single-frequency PPP. To prove the accuracy and the characteristic of real-time time transfer

using BDS-2/3 single-frequency PPP, the mean and STD of the clock difference between BDS-2/3 single-frequency PPP and final products are presented in Figure 23. Similarly, a system bias between BDS-2/3 single-frequency PPP and IGS final precise product is also exists. The STD of the clock difference is 0.65~0.98 ns. Obviously, the accuracy of time transfer using BDS-2/3 single-frequency PPP is poor than that of GPS-only and Galileo. This fact may be caused by two reasons. First, the accuracy of real-time precise products for BDS-2/3 is obviously worse than that of GPS and Galileo (see Figures 2–4). Therefore, we believe that the accuracy of real-time products for BDS-2/3 has room for improvement. Third, nowadays, many stations cannot receive all BDS-3 satellites signals, the real-time precise products used in our work does not include all BDS-3 satellites. Additionally, the pseudorange residuals of BDS-2/3, GPS and Galileo single-frequency PPP with real-time precise products are displayed in Figure 24 for PTBB and HOB2. That can further present the performance of BDS-2/3, GPS and Galileo single-frequency PPP. The RMSs of pseudorange are (0.809, 0.352, 0.614) m and (0.806, 0.246, 0.600) m for BDS-2/3, GPS and Galileo based at PTBB and HOB2 stations. The pseudorange residuals also reflects the correctness of our DCB correction. Hence, the real-time precise products of BDS-2/3 still need to be further improved. In addition, the pseudorange residuals of BDS-2 and BDS-3 satellites at PTBB are listed in Figure 25. obviously, there are no system bias in the pseudorange residuals. Overall, the accuracy of real-time time transfer with BDS-2/3 single-frequency PPP is better than 1 ns at current state. In addition, we believe that with the continuous development of real-time product, we will acquire a better time transfer result using BDS-2/3 single-frequency PPP.

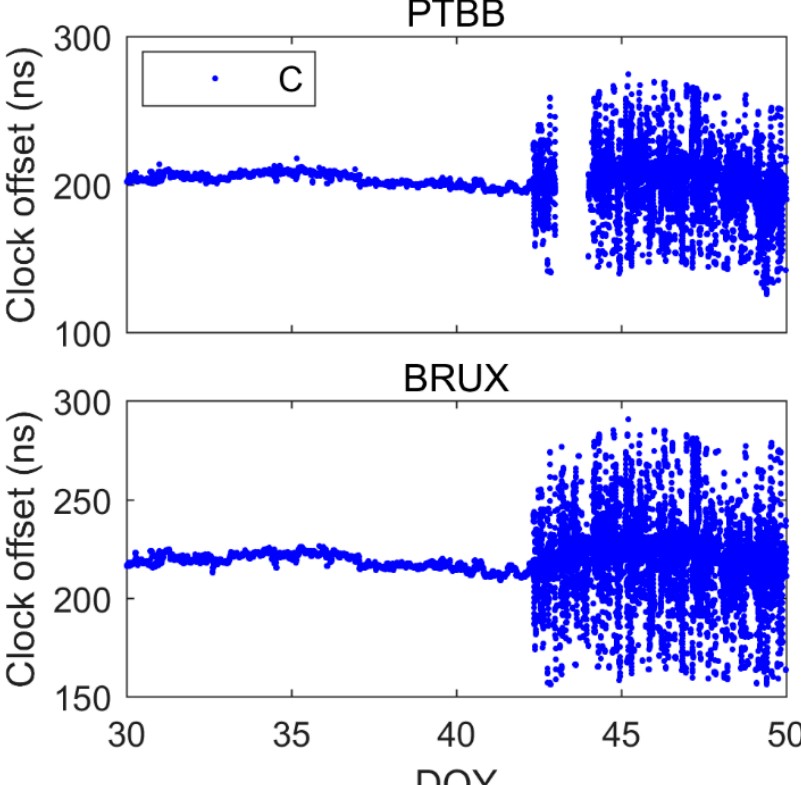

**Figure 20.** Clock offsets of PTBB and BRUX obtained from BDS-2/3 single-frequency PPP. C illustrates BDS-2/3 satellites.

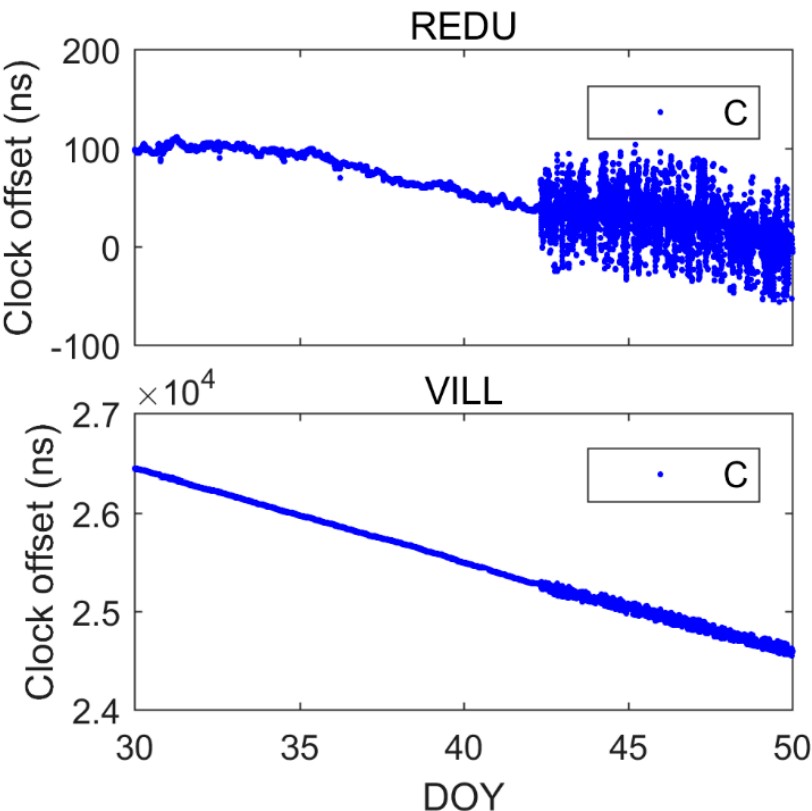

**Figure 21.** Clock offsets of REDU and VILL stations obtained from BDS-2/3-only single-frequency PPP.

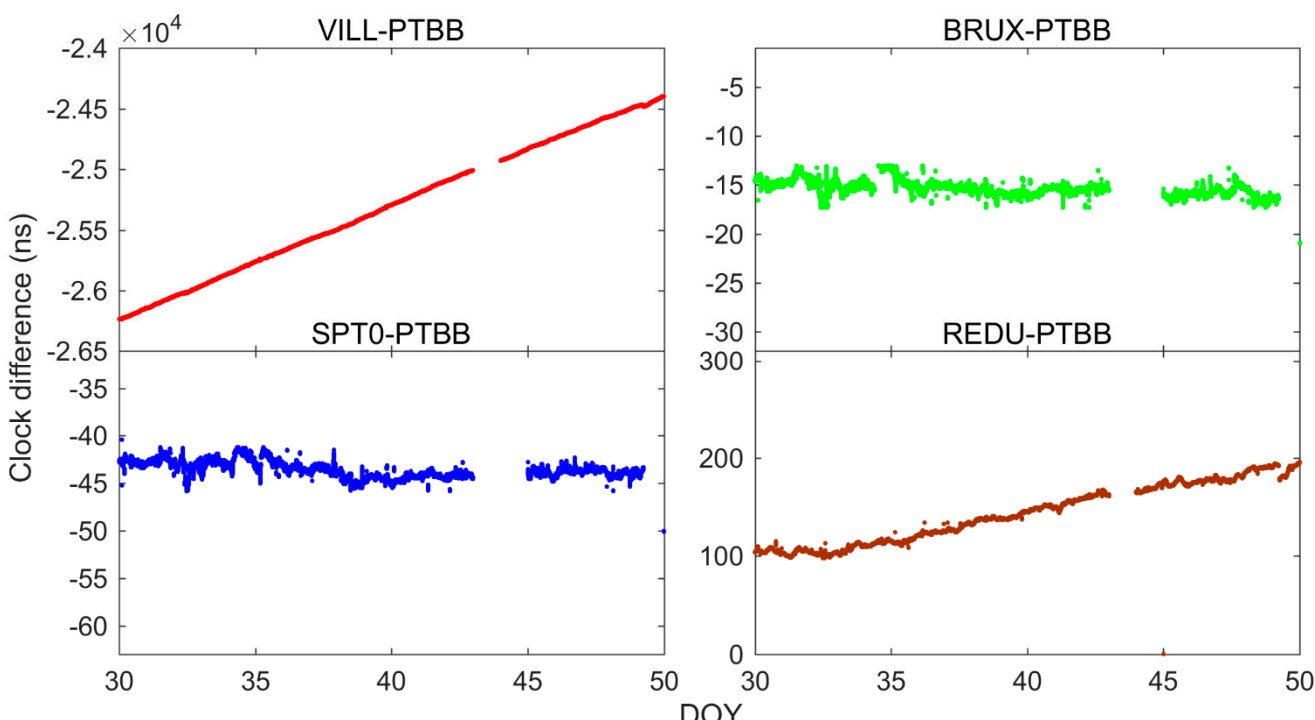

**Figure 22.** Clock difference of VILL-, BRUX-, SPT0- and REDU-PTBB time-links from BDS-2/3-only single-frequency PPP. PT11 has no BDS-3 observations, hence, we use VILL station in this figure.

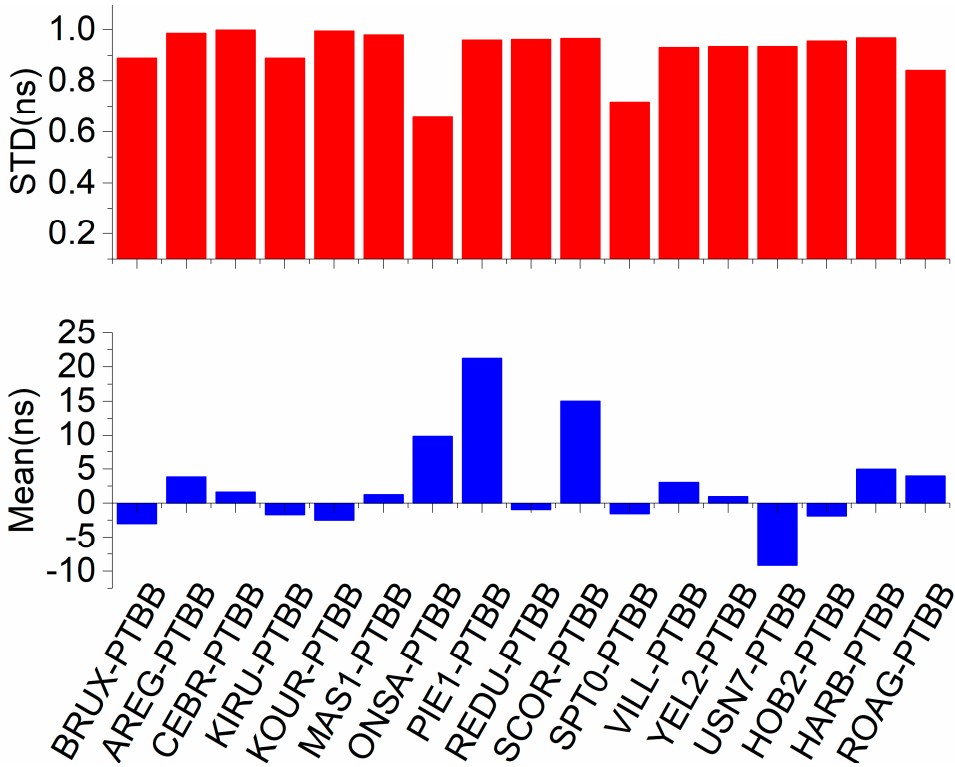

**Figure 23.** The mean and STD values of the clock difference of 18 time-links calculated from BDS-2/3 single-frequency PPP. Note that the clock difference refers to the difference between time transfer solutions from single-frequency PPP and time transfer solutions from IGS final clock products.

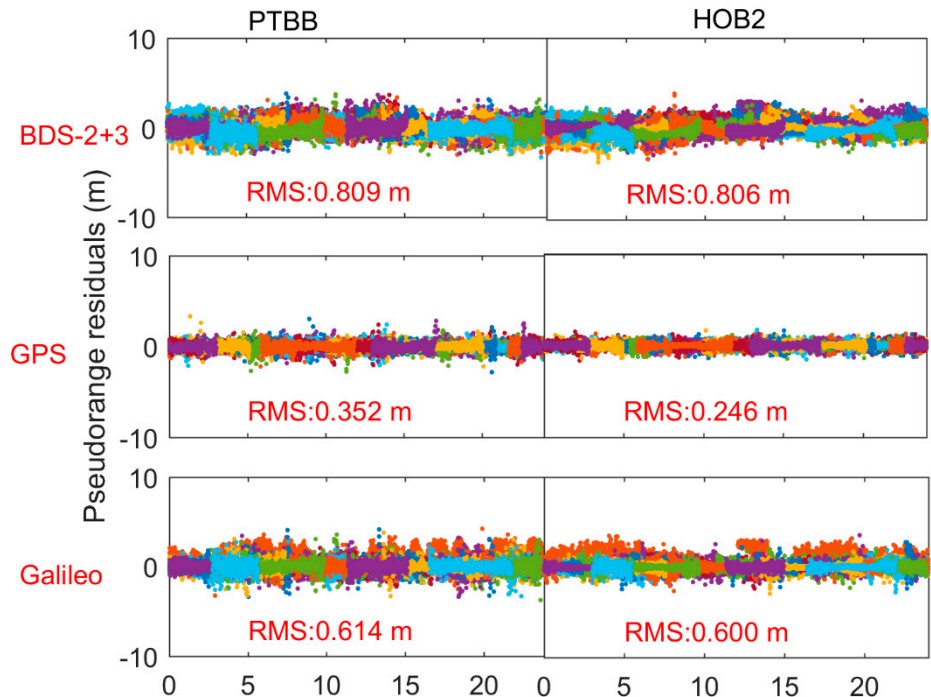

**Figure 24.** The pseudorange residuals of single-frequency PPP with BDS-2/3, GPS-only and Galileo-only satellites.

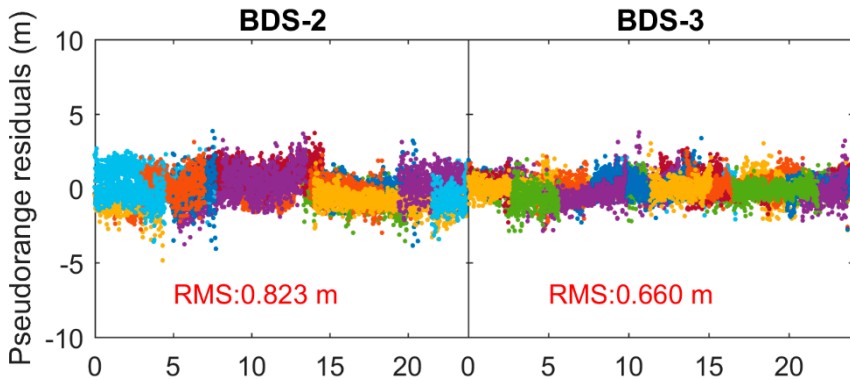

**Figure 25.** The pseudorange residuals of with BDS-2/3 satellites obtained from single-frequency PPP at PTBB station.

Frequency stability is to evaluate BDS-2/3 single-frequency PPP performance in time community from another aspect. Figures 26 and 27 present the MDEV of clock difference obtained from BDS-2/3 single-frequency PPP. The MDEVs of time-links in Figure 25 are (1.3247E-13, 1.9187E-13, 3.3333E-13, 1.3937E-13, 1.9301E-13, 1.4051E-13, 2.5822E-13, 1.9126E-13, 4.7809E-13, 1.7361E-13) at 960 s, respectively, and are (9.652E-14, 1.8415E-13, 2.1716E-13, 8.5925E-14, 2.2849E-13, 9.9375E-14, 2.1474E-13, 9.852E-14, 6.0112E-13, 9.852E-14) at 15,360 s, respectively. In addition, the MDEVs of time-links in Figure 26 are (1.3947E-12, 1.0374E-12, 1.8579E-12, 3.7836E-13, 1.7711E-12, 2.6022E-13, 6.6126E-13) at 960 s and are (3.078E-13, 7.9972E-13, 2.1547E-12, 2.738E-13, 5.3895E-12, 2.2115E-13, 8.324E-13) at 15,360 s, respectively. Overall, the frequency stability obtained from BDS-2/3 single-frequency PPP with real-time precise products can achieve from 1E-12 to 1E-13 level in short-term and 1E-13 level in long-term.

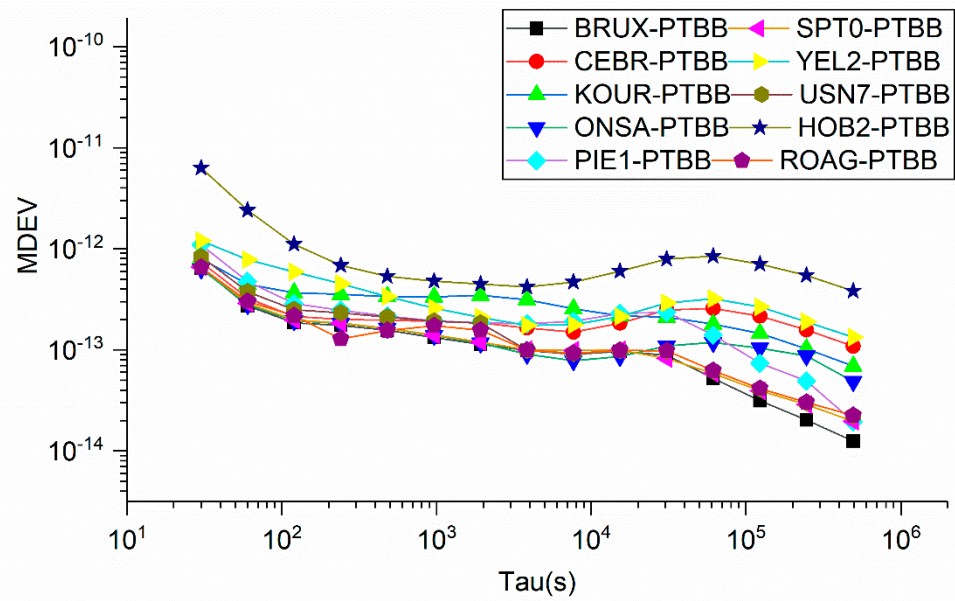

**Figure 26.** MDEV of the time-links obtained from BDS-2/3 single-frequency PPP for 10 stations equipped with H-master clock.

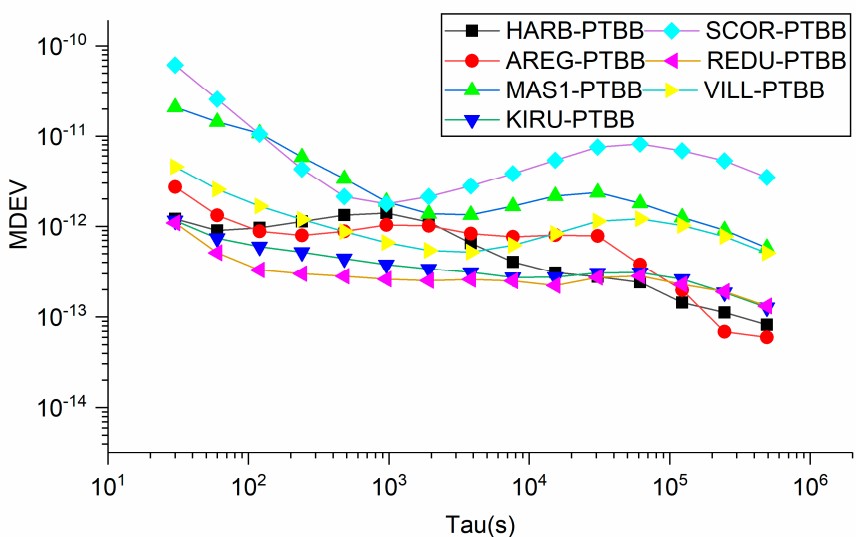

**Figure 27.** MDEV of 7 time-links obtained from BDS-2/3 single-frequency PPP for 7 stations equipped with Rubidium and Cesium clock.

### 5.4. Comparison of BDS-2/3, GPS and Galileo Single-Frequency PPP Time Transfer

In order to compare the performance of real-time single-frequency PPP time transfer with BDS-2/3, GPS and Galileo satellites more clearly. The STD values of the clock difference between time transfer solutions obtained from single-frequency PPP and from IGS final clock products are displayed in Figure 28. Furthermore, MDEVs of the time-links obtained from single-frequency PPP with BDS-2/3, GPS and Galileo satellites at 960 s and 15,360 s are exhibited in Figure 29. Combine Figures 28 and 29, three findings can be concluded. First, the STD values of the clock difference based on BDS-2/3, GPS and Galileo satellites are all better than 1 ns. Second, the STD values of the clock difference from Galileo satellites is better than that of BDS-2/3. The results from GPS outperforms that of Galileo and BDS-2/3 at current statues. Third, for frequency stability at short-term (960 s), MDEVs of the time-links show the similar performance. For frequency stability at long-term (15,360 s), the performance of GPS single-frequency PPP performs best.

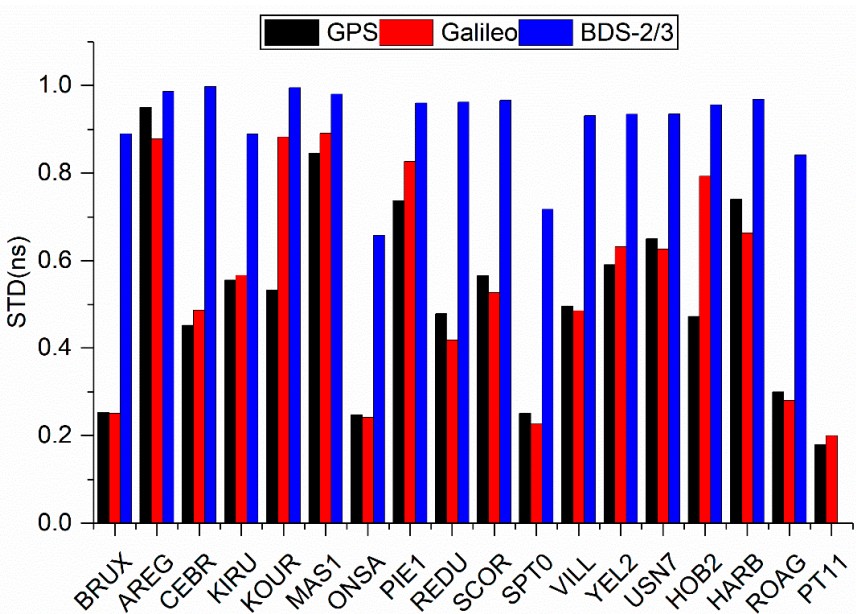

**Figure 28.** The STD values of the clock difference between time transfer solutions from single-frequency PPP with BDS-2/3, GPS and Galileo satellites and IGS final clock products.

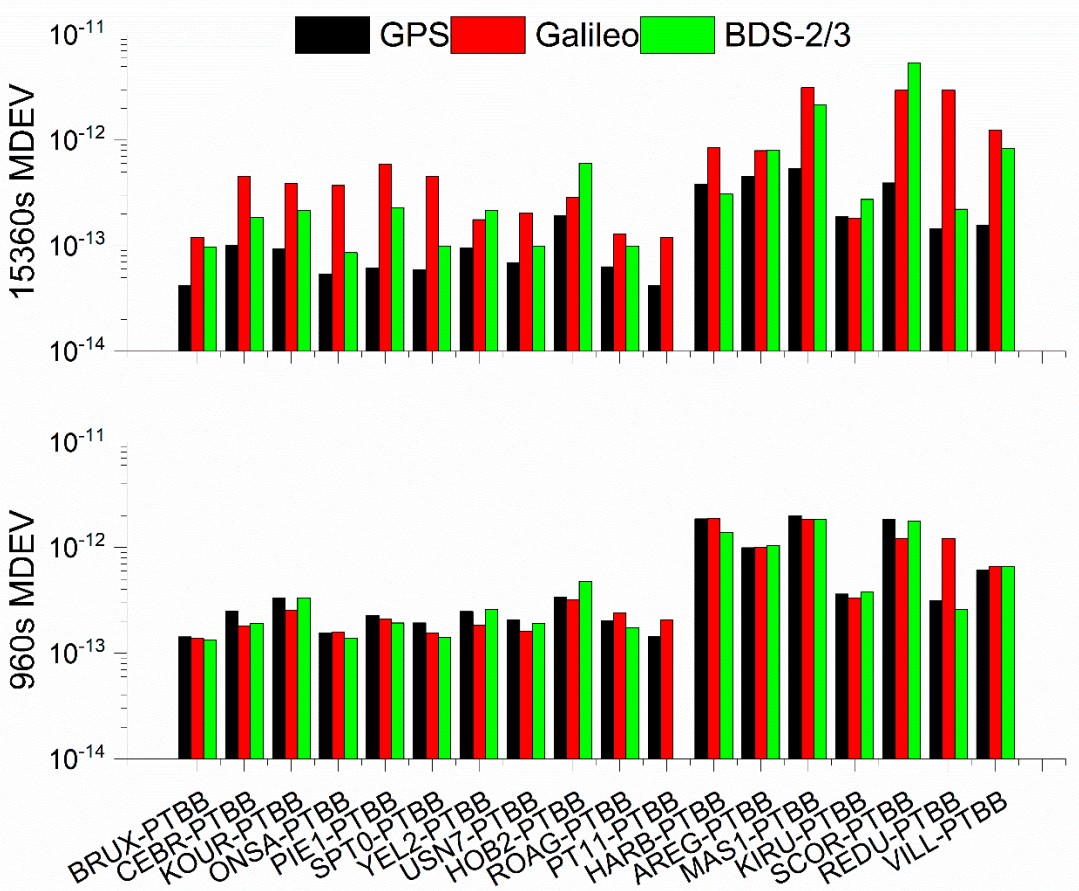

**Figure 29.** MDEV of time-links obtain from single-frequency PPP with BDS-2/3, GPS and Galileo satellites at 960 s and 15,360 s.

## 6. Conclusions

In our work, real-time single-frequency PPP technique is employed to time transfer using BDS-2/3, Galileo- and GPS-only satellites. The investigation of real-time time transfer with single-frequency PPP was studied with 18 MGEX stations and 1 station located in PTB. 20-day observations from DOY 30 to 50, 2021 were selected and tested. The un-combine single-frequency PPP model with ionospheric-constraint was used in this study. The real-time products were evaluated firstly. Then, real-time time transfer using BDS-2/3, GPS- and Galileo-only single-frequency PPP were investigated with corresponding real-time products. Two findings are concluded as follow:

First, RMSs of orbits errors are (0.026, 0.038, 0.032) and (0.038, 0.080, 0.051) m for GPS and Galileo at RAC direction. RMSs of orbits for BDS-2 GEO, IGSO/MEO and BDS-3 are (0.35, 3.73, 1.84), (0.16, 0.32, 0.19) and (0.11, 0.28, 0.16) m. The mean STD values are (0.18, 0.16) ns for GPS and Galileo. Furthermore, the mean STD values for BDS-2 GEO, BDS-2 IGSO/MEO and BDS-3 satellites are (0.9, 0.58, 0.51) ns, respectively.

Second, real-time single-frequency PPP can be used for time transfer. The mean STD values of clock difference are (0.51, 0.54, 0.91) ns obtained from GPS-only, Galileo-only and BDS-2/3 single-frequency PPP with real-time precise products at current state. The short-term frequency stability of GPS-only, Galileo-only and BDS-2/3 single-frequency PPP achieve (1E-13, 1E-13, 1E-12) level, respectively. In addition, the frequency stability of GPS-only, Galileo-only and BDS-2/3 single-frequency PPP will reach (1E-14, 1E-13, 1E-14) level in long-term.

With the continuous development of real-time products of BDS, we expect BDS-3 single-frequency PPP time transfer to achieve better accuracy.

**Author Contributions:** X.X. and F.S. designed the experiments; the paper was wrote by X.X.; F.S., X.L., P.S. and Y.G. modified this paper. All authors have read and agreed to the published version of the manuscript.

**Funding:** This contribution was supported by Natural Science Foundation of Jiangsu Province (No. BK20190714; BK20201374), the National Natural Science Foundation of China (No. 41904018; 42077003; 42104014), K. C. Wong Education Foundation, High-level innovation and entrepreneurship talent plan of Jiangsu Province and the West Light Foundation of The Chines Academy of Sciences (XAB2018B20).

**Institutional Review Board Statement:** Not applicable.

**Informed Consent Statement:** Not applicable.

**Data Availability Statement:** The datasets used in this work are provided by PTB and IGS.

**Acknowledgments:** The authors acknowledge IGS for providing observations, and CNES for real-time products.

**Conflicts of Interest:** The authors declare no conflict of interest.

## Abbreviation

| | |
|---|---|
| AV | All-in-view |
| BDS | BeiDou Navigation Satellite System |
| A | Along |
| R | Radio |
| C | Cross |
| CNES | Centre National d'Etudes Spatiales |
| CV | Common-view |
| DCB | Differential code bias |
| DOY | Day of year |
| GNSS | Global navigation satellite system |
| GEO | Geostationary earth orbit |
| GLONASS | GLObal NAvigation Satellite System |
| IGSO | Inclined geosynchronous orbit |
| TAI | International Atomic Time |
| IFCB | Inter-frequency code biases |
| IPPP | Integer ambiguity PPP techniques |
| IPP | Ionospheric piece point |
| IGS | International GNSS service |
| ISB | Inter system bias |
| OMC | Observed-minus-computed |
| MEO | Medium earth orbit |
| MDEV | Modified Allan deviation |
| MGEX | Multi-GNSS experiment |
| PCO | Phase center offsets |
| PCV | Phase center variations |
| RMS | Root mean squares |
| SSR | State Space Representation |
| STD | Standard deviation |
| 3D | Three dimensions |
| TF | Triple-frequency |
| UCD | Uncalibrated code bias |
| ZWD | Zenith wet delay |

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
