# Peer review of "Performance of BDS-2/3, GPS, and Galileo Time Transfer with Real-Time Single-Frequency Precise Point Positioning"

_remotesensing, doi:10.3390/rs13214192_

Round 1
Reviewer 1 Report
The study tries to evaluation the performance of real-time GNSS time transfer using single frequency precise point positioning. Although the topic is interesting and the authors have done a lot of data analysis, the presentation is rather poor in both the manuscript organization and the language. The major comments are addressed here and the special comments are annotated in the uploaded file.
Major comments:
- First of all, what are the most critical aspects for time transfer using PPP? This do not seem to be clearly addressed in this study. In the assessment, we noticed the estimated clocks could have very large bias, up to 200 ns (Fig.6) and the bias may change from epoch to epoch up to 50 ns. Why these biases do not matter?
- One of the major issues using single frequency data is the ionospheric delay. The authors claimed the DESIGN model (Eq (3)) is employed, but in Eq (4) slant ionospheric delays are estimated. More important is that the impact of the ionospheric delay modelling/parameterization has not been investigated in the study. For example, whether and to what extent the estimated ionospheric delays agree with the “true”, for example that extracted from double-frequency data.
- In principle, ranges define the reference of the clock estimates, while phases decide the stability. Range biases at receiver- and satellite-end are very critical in time transfer. It is already confirmed that there is a range bias at the receiver-end between BDS2 and 3 satellites (similar to the inter-system range bias at the receiver-end). It is not clearly stated how this bias is handled and if the large range residuals for BDS2/3 shown in Fig. 24 is correlated to this range bias?
- The estimated clocks have very large scatter after day-of-year 42, 2021 and the large scatter vanished in the clock-difference between stations. I agree with the explanation that is the problem in the products. However, is there any way to confirm this fact and what kind of products should be provided for precise time transfer? This could be an interesting point of this study?
- Although the language is in general understandable, it is wordy and there are many grammar mistakes. The presentations of the result of GPS, Galileo and BDS2/3 are almost the same, as the results are very similar. These should be summarized together and better with a Table of all the statistics instead of listed the values in the text.

Author Response
Thank you very much for your encouragement and comments. We have revised the manuscript carefully and responded point by point to the comments as below. (C and R indicate comment and response, respectively). Our revisions are highlighted in our manuscript using the "Track Changes" function.
C:First of all, what are the most critical aspects for time transfer using PPP? This do not seem to be clearly addressed in this study. In the assessment, we noticed the estimated clocks could have very large bias, up to 200 ns (Fig.6) and the bias may change from epoch to epoch up to 50 ns. Why these biases do not matter?
R:Time transfer solutions is to obtain the clock difference between different stations. In fig.6, the receiver clock offset is the difference between the reference time of precise product and the local time. The large bias is affected by the reference of precise product. But the reference of precise product will be eliminated completely for time transfer, see eqs. 5 and 6
C: One of the major issues using single frequency data is the ionospheric delay. The authors claimed the DESIGN model (Eq (3)) is employed, but in Eq (4) slant ionospheric delays are estimated. More important is that the impact of the ionospheric delay modelling/parameterization has not been investigated in the study. For example, whether and to what extent the estimated ionospheric delays agree with the “true”, for example that extracted from double-frequency data.
R: we are sorry for that mistakes, we have delete Eq(4). In our work, in to extract the receiver clock offset, the slant ionospheric delays obtained from GIM model were applied to constraint the single-frequency PPP with DESIGN model.
C: In principle, ranges define the reference of the clock estimates, while phases decide the stability. Range biases at receiver- and satellite-end are very critical in time transfer. It is already confirmed that there is a range bias at the receiver-end between BDS2 and 3 satellites (similar to the inter-system range bias at the receiver-end). It is not clearly stated how this bias is handled and if the large range residuals for BDS2/3 shown in Fig. 24 is correlated to this range bias?
R: Thanks, the ISB between BDS-2 and BDS-3 was estimated as white noise, we have added it in the revision. And we have added fig.25 which exhibit the residuals of BDS-2 and BDS-3, respectively.
C: The estimated clocks have very large scatter after day-of-year 42, 2021 and the large scatter vanished in the clock-difference between stations. I agree with the explanation that is the problem in the products. However, is there any way to confirm this fact and what kind of products should be provided for precise time transfer? This could be an interesting point of this study?
R: the large bias has been pointed out at “Ge, Y.; Qin, W.; Su, K.; Yang, X.; Ouyang, M.; Zhou, F.; Zhao, X. A new approach to real-time precise point-positioning timing with International GNSS Service real-time service products. Measurement Science and Technology 2019, 30, 125104, doi:10.1088/1361-6501/ab2fa5.” Additionally, single-frequency PPP time transfer with IGS final products or GBM product has no that larger bias. No matter the reference of the product changes or fluctuates, the time transfer will not be affected and the reference will be eliminated, as can be seen from equation (5) - (6).
C: Although the language is in general understandable, it is wordy and there are many grammar mistakes. The presentations of the result of GPS, Galileo and BDS2/3 are almost the same, as the results are very similar. These should be summarized together and better with a Table of all the statistics instead of listed the values in the text.
R: thanks, we have modified the grammar. And we have added the comparison of GPS, Galileo and BDS-2/3 single-frequency PPP time transfer for clearly presentation at subsection 5.4.
R: Thank you for your all comments in the attachment file, we have modified all the comments according to your suggestion.
Reviewer 2 Report
GENERAL REMARKS:
The paper studies the possibility of time transfer using only a single frequency GNSS receiver. This is done with the analysis of the data provided by 18 MGEX stations. I am missing this information in the abstract, since it describes what has really been done.
I am also missing a list of used MGEX stations (although it could be deduced from eg. Figs 11 and 12), their location and the hardware they use.
As you said, the motivation for this analysis was the availability and the price of the single frequency receivers. However, an important thing that should be mentioned at least, is that the data those receivers would provide, would be of much worse quality with respect to the data provided by MGEX stations. Furthermore, a lot of those cheap receivers lack the phase calculations ability, although you probably didn't have those in mind.
Nevertheless, I find the results of this research interesting in the sense that the results present what can be expected if such an approach is used.
PARTICULAR REMARKS (grammatical errors are not included):
1. There is another thing that bothers me in the abstract - the presented numbers probably present the relative frequency stability (probably, there is also a minus sign missing).
2. The table of abbreviations is not consistent - the position of the abbreviations and their descriptions change. Then CENS and CNES are used interchangeably within the text.
3. -10-16/T should be described more verbosely.
4. UTC(K) is not defined.
5. Section 2.1. is the weakest part of the paper. Equation 1 is inconsistent and not all quantities are well defined:
- The terms don't have the same physical unit (eg. dt presents time, \Delta x displacement - probably there is the speed of light missing).
- p and l are badly defined - OMC of what? - regardless of the fact that from the context it can be deduced that those are probably the pseudorange and phase.
- What is the meaning of indices s and r? Is it GNSS used and receiver?
- Where does the index j step in? What does it mean? Frequency of what?
- e is the unit vector in what direction?
- DCB is defined as a difference between two d^s_i. However, the latter are not defined.
6. No matter how hard I try to combine those equations in a matrix form, I cannot reconstruct equation 4.
7. The text in section 2.2. is inconsistent with Fig. 1. - Broadcast ephemeris are obtained from the station.
8. Fig.3 is split for BDS 2 and 3. A word of which belongs to what would improve the clarity of the figure.
9. Why was PTBB chosen as the reference station?
10. What was the motivation (in Figs. 6-8) to show those particular stations. Why were they grouped by two? What does G mean (OK, it can be deduced that it stands for GPS from the further reading, but it is completely redundant). What about the other stations. Do they exhibit a similar behavior to those. I would strongly suggest an online repository with data for all the stations (with a link in the references at the end of the article).
11. Page 9, second paragraph: The argumentation of the reason of increased fluctuations after DOY 43 is weak.
12. Page 10, last paragraph: what do you mean by system difference? Is it systematic difference, hardware induced difference or something else?
13. Page 11, first paragraph: REDU-PTBB has a shape of a linearly increasing function. However, it is not necessary that it will be linear over a longer period - after all, it cannot increase indefinitely.
14. Page 11, par. 1: Fig. 10 is not explained well. For instance, STD of REDU should be substantially bigger than that. Or is it a short term deviation? I am even more confused on what is the mean value it represents.
15. Fig. 9. I suggest to change the color of REDU-PTBB. It doesn't print well in black and white.
16. I find it very interesting, that the fluctuations after DOY 43 in clock offset of eg. PTBB and BRUX are of order of 100ns. Their difference, on the other hand, is only of order of 1ns. I am missing some analysis here. Maybe it could give you a better explanation for the reason of the former.
17. Maybe it should be mentioned in the caption of figure 10 that those values are relative to PTBB.
18. The definition of MDEV is missing.
19. The definition of tau is missing.
20. The same remarks as in section 5.1. apply also to sections 5.2. and 5.3.
21. What can be deduced from the fact that beside the fluctuations (and some isolated points in the Galileo's case) the GPS and Galileo provide a practically identical plot?
22. The remark above applies to BDS to some extent. However, there is a small yet notable difference that brings me in disagreement with your conclusions in the beginning of the section 5.3.: while the clock offsets are rather constant before DOY 43 in GPS and Galileo case, BDS shows a slight parabolic behavior (see eg. Fig. 20). Could it indicate a systematic error in the system of satellite's clocks in the BDS' case?
23. Page 18, line 11: Fig. 23 should probably be Fig. 24.
24. I find it somewhat disturbing that the station gets changed (VILL in this case) without providing a reason for that.
25. The only informative thing in Fig. 24 are the RMS values. I cannot deduce anything useful form those plots.
Author Response
Thank you very much for your encouragement and comments. We have revised the manuscript carefully and responded point by point to the comments as below. (C and R indicate comment and response, respectively). Our revisions are highlighted in our manuscript using the "Track Changes" function.
C: The paper studies the possibility of time transfer using only a single frequency GNSS receiver. This is done with the analysis of the data provided by 18 MGEX stations. I am missing this information in the abstract, since it describes what has really been done.
R: Thanks, we have added it.
C: I am also missing a list of used MGEX stations (although it could be deduced from eg. Figs 11 and 12), their location and the hardware they use.
R: Thanks, we have added it in the revision.
C: As you said, the motivation for this analysis was the availability and the price of the single frequency receivers. However, an important thing that should be mentioned at least, is that the data those receivers would provide, would be of much worse quality with respect to the data provided by MGEX stations. Furthermore, a lot of those cheap receivers lack the phase calculations ability, although you probably didn't have those in mind.
R: Thank you for your suggestions. We have not obtained the data of the single-frequency receiver of the external atomic clock. In the future work, we will purchase relevant equipment to do experiments.
C: There is another thing that bothers me in the abstract - the presented numbers probably present the relative frequency stability (probably, there is also a minus sign missing).
R: Thanks, we have presented the frequency stability of time transfer solution. And we have modified in the revision.
C: The table of abbreviations is not consistent - the position of the abbreviations and their descriptions change. Then CENS and CNES are used interchangeably within the text.
R: Accepted and modified.
C: -10-16/T should be described more verbosely.
R: Thanks, T is the duration in days of continuous phase measurements. We have modified it.
C: UTC(K) is not defined.
R: Thanks. That is UTC(k), where k represents the time lab.
R: The terms don't have the same physical unit (eg. dt presents time, \Delta x displacement - probably there is the speed of light missing).
C: We have added it in the revision.
C: p and l are badly defined - OMC of what? - regardless of the fact that from the context it can be deduced that those are probably the pseudorange and phase.
R: Yes, p and l are the OMC of the pseudorange and phase, we have modified it.
C: What is the meaning of indices s and r? Is it GNSS used and receiver?
R: yes, s and s refer to satellite and receiver, respectively. We have added it.
C: Where does the index j step in? What does it mean? Frequency of what?
R: j is the second frequency.
C: e is the unit vector in what direction?
R: e is the unit vector of the component from the receiver to the satellites
C: DCB is defined as a difference between two d^s_i. However, the latter are not defined.
R: Thanks, we have added it.
C: No matter how hard I try to combine those equations in a matrix form, I cannot reconstruct equation 4.
R: We are sorry for this mistake. We have deleted it.
C: The text in section 2.2. is inconsistent with Fig. 1. - Broadcast ephemeris are obtained from the station.
R: Thanks, we have modified it.
C: Why was PTBB chosen as the reference station?
R: Because PTBB located at PTB (Physikalisch-Technische Bundesanstalt). In addition, PTBB connected to UTC(PTB).
C: What was the motivation (in Figs. 6-8) to show those particular stations. Why were they grouped by two? What does G mean (OK, it can be deduced that it stands for GPS from the further reading, but it is completely redundant). What about the other stations. Do they exhibit a similar behavior to those. I would strongly suggest an online repository with data for all the stations (with a link in the references at the end of the article).
R: Thanks, the stations in Fig. 6-8 are selected randomly, we can not exhibit all the results. In addition, the observations and precise products in our work can be downloaded from IGS and CNES, everyone can obtain them from the internet for free.
C: Page 9, second paragraph: The argumentation of the reason of increased fluctuations after DOY 43 is weak.
R: Thanks, the fluctuations mainly are affected by the reference of the real-time products. It has been proved by Ge et al.
Ge, Y.; Qin, W.; Su, K.; Yang, X.; Ouyang, M.; Zhou, F.; Zhao, X. A new approach to real-time precise point-positioning timing with International GNSS Service real-time service products. Measurement Science and Technology 2019, 30, 125104, doi:10.1088/1361-6501/ab2fa5.
C: Page 10, last paragraph: what do you mean by system difference? Is it systematic difference, hardware induced difference or something else?
R: The system difference is the difference between time transfer solutions from single-frequency PPP and IGS final clock product.
C: Page 11, first paragraph: REDU-PTBB has a shape of a linearly increasing function. However, it is not necessary that it will be linear over a longer period - after all, it cannot increase indefinitely.
R: The shape of a linearly increasing is determined by the characteristic of the atomic for REDU.
C: Page 11, par. 1: Fig. 10 is not explained well. For instance, STD of REDU should be substantially bigger than that. Or is it a short term deviation? I am even more confused on what is the mean value it represents.
R: as we pointed out, the clock difference is the difference between time transfer solutions from single-frequency PPP and IGS final clock product. So the trend of the time series has been removed. Hence, the STD values of the clock difference is a long-term deviation. And we have added the explanation in the revision.
C: Fig. 9. I suggest to change the color of REDU-PTBB. It doesn't print well in black and white.
R: Accepted and modified.
C: I find it very interesting, that the fluctuations after DOY 43 in clock offset of eg. PTBB and BRUX are of order of 100ns. Their difference, on the other hand, is only of order of 1ns. I am missing some analysis here. Maybe it could give you a better explanation for the reason of the former.
R: For the receiver clock offset at PTBB and BRUX station is the difference between the reference time of precise product and the local time. Hence, the fluctuations of clock offset after DOY 43 are mainly affected by the reference time of precise product. However, the time transfer solutions will eliminate the reference time of product, the solutions only is difference between the time of PTBB and BRUX. We have added it in the revision.
C: Maybe it should be mentioned in the caption of figure 10 that those values are relative to PTBB.
R: accepted and modified
C: The definition of MDEV is missing.
R: accepted and modified.
C: The definition of tau is missing.
R: Tau is the sampling period. We have added it.
C: The same remarks as in section 5.1. apply also to sections 5.2. and 5.3.
R: accepted and modified.
C: What can be deduced from the fact that beside the fluctuations (and some isolated points in the Galileo's case) the GPS and Galileo provide a practically identical plot?
R: GPS L1 and Galileo E1 present the same frequency. Hence, the figures for GPS and Galileo show the similar performance. We want to introduce the performance of GPS or Galileo single-frequency PPP time transfer, respectively.
C: The remark above applies to BDS to some extent. However, there is a small yet notable difference that brings me in disagreement with your conclusions in the beginning of the section 5.3.: while the clock offsets are rather constant before DOY 43 in GPS and Galileo case, BDS shows a slight parabolic behavior (see eg. Fig. 20). Could it indicate a systematic error in the system of satellite's clocks in the BDS' case?
R: Thanks, from Fig. 20 and 21, a slight parabolic behavior for BDS-2/3 is mainly affected by the reference time of real-time products.
C: Page 18, line 11: Fig. 23 should probably be Fig. 24.
R: accepted and modified.
C: I find it somewhat disturbing that the station gets changed (VILL in this case) without providing a reason for that.
R: thanks, because PT11 have no BDS-3 observations, we selected VILL station randomly for presentation. We have added the explanation in the revision.
C: The only informative thing in Fig. 24 are the RMS values. I cannot deduce anything useful form those plots.
R: for fig. 24, the RMS values of residuals can also evaluate the performance of single-frequency PPP time transfer. Additionally, there is no obviously system bias, hence, the DCB correction is right.
Round 2
Reviewer 1 Report
The manuscript is well revised and all my questions are clarified. However, the large noise in the estimated clocks on the second half of the experimental data should be confirmed by computing the differences between the reference (IGS final) and the CNES real-time products. The epoch-wise average over satellites should be "clock reference difference" which should vary very much from epoch to epoch in the 2nd half time. This is also why two stations are needed for high-precision time transfer which should be also explained in the paper.
Anyway, the above-mentioned aspects might be commonsense for experts in time transfer but not for all reader working on GNSS. To make them clear would be a great help for most fo the readers.
Author Response
The manuscript is well revised and all my questions are clarified. However, the large noise in the estimated clocks on the second half of the experimental data should be confirmed by computing the differences between the reference (IGS final) and the CNES real-time products. The epoch-wise average over satellites should be "clock reference difference" which should vary very much from epoch to epoch in the 2nd half time. This is also why two stations are needed for high-precision time transfer which should be also explained in the paper.
Anyway, the above-mentioned aspects might be common sense for experts in time transfer but not for all reader working on GNSS. To make them clear would be a great help for most fo the readers.
R: Thanks, we have added the results obtained from single-frequency PPP with IGS final products in Fig. 6. The results show a stable time series, which prove our findings.
Reviewer 2 Report
The modifications improved the readability and clarity of the paper. The authors took into account the majority of my previous remarks. However, some of them were not addressed enough. Some of remarks are intended to improve the quality of the paper to be acceptable to a broader audience. Yet, there are still two points in my opinion (namely, remarks 5 and 11) that should be improved before the paper is published.
ad General Remarks: A list of receivers (perhaps a non-exhausting one) that take phase calculations into account, use a single frequency approach and are in the lower price range would give your motivation a more solid foundation.
ad 1: I am still missing the word "relative". It is about relative stability.
ad 2: I probably wasn't clear enough. The thing that bothered me the most is that in some cases the abbreviations are on the left side of the column and on the right in the other cases. It is confusing.
ad 4: Thank you for the information, but it should still be explained in the text.
ad 5: The equations description has improved. Although I am still confused about the meaning of index j, since you are using a single frequency approach.
ad 11: It is pretty much clear that the fluctuations are affected by the reference of the real-time products (that was where the hint in point 16 was pointing to). However, the explanation is still weak. It is obvious that something significantly happened on DOY 43. And this cannot be attributed only to fluctuations.
ad 12: I still think this description should be put in the text.
Author Response
ad General Remarks: A list of receivers (perhaps a non-exhausting one) that take phase calculations into account, use a single frequency approach and are in the lower price range would give your motivation a more solid foundation.
R: Thanks for your suggestions, your comments are very good. But there are no single-frequency receiver equipped with high-precise atomic clock.
ad 1: I am still missing the word "relative". It is about relative stability.
R: Thanks, we want to say that the time series of BRUX or are more stable than that of REDU VILL. Now, we have deleted the word “relative”
ad 2: I probably wasn't clear enough. The thing that bothered me the most is that in some cases the abbreviations are on the left side of the column and on the right in the other cases. It is confusing.
R: We are sorry for those mistakes. We have modified it in the revision.
ad 4: Thank you for the information, but it should still be explained in the text.
R: Thanks, we have added it in the revision.
ad 5: The equations description has improved. Although I am still confused about the meaning of index j, since you are using a single frequency approach.
R: Thanks, we just use the pseudorange and phase observations at frequency I, but frequency j was just used to calculated the corresponding coefficient.
ad 11: It is pretty much clear that the fluctuations are affected by the reference of the real-time products (that was where the hint in point 16 was pointing to). However, the explanation is still weak. It is obvious that something significantly happened on DOY 43. And this cannot be attributed only to fluctuations.
R: Thanks, we have added the single-frequency PPP with IGS final products. The results show that the time series are very stable, which can prove our conclusions.
ad 12: I still think this description should be put in the text.
R: Thanks, we have added it.